# CultureLLM: Incorporating Cultural Differences into Large Language Models

**Cheng Li**[*]
Institute of Software, CAS
chenglicat0228@gmail.com

**Mengzhuo Chen**
Institute of Software, CAS
mengzhuo.happy@gmail.com

**Jindong Wang**[†]
Microsoft Research
jindong.wang@microsoft.com

**Sunayana Sitaram**
Microsoft Research
Sunayana.Sitaram@microsoft.com

**Xing Xie**
Microsoft Research
xing.xie@microsoft.com

## Abstract

Large language models (LLMs) have been observed to exhibit bias towards certain cultures due to the predominance of training data obtained from English corpora. Considering that multilingual cultural data is often expensive to procure, existing methodologies address this challenge through prompt engineering or culture-specific pre-training. However, these strategies may neglect the knowledge deficiency of low-resource cultures and necessitate substantial computing resources. In this paper, we propose **CultureLLM**, a cost-effective solution to integrate cultural differences into LLMs. CultureLLM employs the World Value Survey (WVS) as seed data and generates semantically equivalent training data through the proposed semantic data augmentation. Utilizing only 50 seed samples from WVS with augmented data, we fine-tune culture-specific LLMs as well as a unified model (CultureLLM-One) for 9 cultures, encompassing both rich and low-resource languages. Extensive experiments conducted on 60 culture-related datasets reveal that CultureLLM significantly surpasses various counterparts such as GPT-3.5 (by 8.1%) and Gemini Pro (by 9.5%), demonstrating performance comparable to or exceeding that of GPT-4. Our human study indicates that the generated samples maintain semantic equivalence to the original samples, offering an effective solution for LLMs augmentation. Code is released at https://github.com/Scarelette/CultureLLM.

## 1  Introduction

Culture is a complex construct that encapsulates various identities, including, but not limited to, language, nationality, region, religion, and gender identity. Cultural bias is prevalent worldwide and refers to the tendency to favor specific cultural perspectives, values, and norms, which results in subjective opinions and may offend individuals from other cultures. For instance, according to the World Value Survey [Survey, 2022], Arabic culture believes that men are better political leaders than

---

[*]Work done during Cheng's internship at MSRA.
[†]Corresponding author.

38th Conference on Neural Information Processing Systems (NeurIPS 2024).

women, while people in the United States maintain a contrary opinion.[3] As large language models (LLMs) [OpenAI, 2023b, Google, 2023] gain prominence, they are reported to exhibit cultural bias and specifically show partiality towards Western culture, as English corpora dominate the training data [Liu et al., 2023b, Cao et al., 2023, Masoud et al., 2023, Naous et al., 2023, Wang et al., 2023d, Johnson et al., 2022]. Low-resource cultures are frequently underrepresented due to the insufficient training data available from these cultures. LLMs' cultural bias constitutes a significant bottleneck in human-AI collaboration and considerably impedes AI democracy.

Tackling cultural bias necessitates that a large language model (LLM) acknowledges cultural differences [Hofstede, 1984]. Kovavc et al. Kovač et al. [2023] and Want et al. Wang et al. [2023d] thought LLMs have enough knowledge of all cultures and devised prompt engineering technologies to induce LLMs to exhibit specific cultural perspectives. However, they are not effective, especially in low-resource cultures with limited data. Another line of work pre-trained culturally aware LLMs and then fine-tuned on specific datasets [Chan et al., 2023, Nguyen et al., 2023b, Pipatanakul et al., 2023, Abbasi et al., 2023, Lin and Chen, 2023]. They require the collection of large-scale pre-training and fine-tuning datasets and extensive computing resources, thus are not affordable to ordinary researchers and cannot handle low-resource culture. To date, training culturally aware LLMs at affordable costs remains a challenge.

In this paper, we propose **CultureLLM**, a cost-effective[4] solution to incorporate cultural differences into LLMs. Technically speaking, CultureLLM is inspired by the well-known fact that LLMs are inevitably not robust to the style and format of the prompts [Zhu et al., 2023], indicating that we can further leverage such a weakness to further improve the performance of LLMs by enriching the prompts. In particular, we focus on cultural values in this work. As shown in Figure 1, CultureLLM consists of three steps: sampling, semantic data augmentation, and fine-tuning. Inspired by Attitude-Behavior Consistency theory [Fazio and Zanna, 1981] which emphasizes that people's opinion is consistent with their behaviors, we use the World Values Survey (WVS) [Survey, 2022] as seed data. Then, we devise a semantic data augmentation approach to generate semantically equivalent samples. The aim is to generate semantic equivalent inputs,

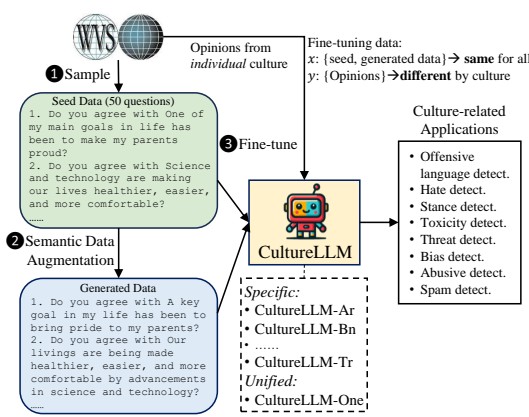

Figure 1: Overview of CultureLLM. CultureLLM consists of three steps: sampling, semantic data augmentation, and fine-tuning. Culture-specific and unified CultureLLM can be fine-tuned.

thus we can get the ground-truth from seed data directly. Finally, CultureLLM is obtained by fine-tuning both the seed and the generated data. WVS is a public opinion poll that contains people's opinions on cultural topics from different countries. To be specific, we select 50 seed samples from WVS, covering 7 topics: "social values", "migration", "security", "science and technology", "religious values", "ethical values and norms", and "political interest and political participation". Using these generated samples and answers from people in different cultures, we fine-tune specific and unified LLMs: specific LLMs are tailored for each culture such as CultureLLM-Ar for Arabic and CultureLLM-Tr for Turkish; unified LLMs (CultureLLM-One) are one LLM that fits all cultures.[5]

We build 9 specific CultureLLM and a CultureLLM-One covering both high- and low-resource cultures: Arabic culture, Bengali culture, Chinese culture, English culture, German culture, Korean culture, Portuguese culture, Spanish culture, and Turkish culture. Then we evaluated them on 8 culture-related downstream tasks: offensive language detection, hate speech detection, stance detection, toxicity detection, threat detection, bias detection, abusive detection, spam detection, and an open-ended generative task. We have 60 test sets of 68, 672 samples in total. Experiments show that CultureLLM fine-tuned on GPT-3.5 significantly outperforms GPT-3.5 by 8.1% and outperforms Gemini pro [Google, 2023] by 9.5% on average F1 score, achieving comparable or even better performance with GPT-4. Our human study of 50 people demonstrates that the augmentation

---

[3]We respect all opinions in different cultures.

[4]Fine-tuning a CultureLLM only costs $6 via OpenAI API.

[5]The bound of culture is unclear; we use the main spoken language to distinguish cultures [Delanoy, 2020b].

method can generate semantically equivalent samples. We further interpret the rationale behind its effectiveness by exploring the fine-tuning data size and case studies. Finally, results on Big-Bench Hard [Suzgun et al., 2022] and GSM8K [Cobbe et al., 2021] indicate that CultureLLM is resistant to catastrophic forgetting. CultureLLM also supports fine-tuning LLMs of open-source models.

Our contributions are three-fold:

1. We presented CultureLLM, a cost-effective fine-tuning solution to build culturally-aware LLMs.
2. We proposed semantic data augmentation, an augmentation approach to generate high-quality and diverse training data for LLMs.
3. We conducted extensive experiments in a wide range of cultures and LLMs, showing that LLMs performs consistently well in all downstream tasks.

## 2    Related Work

### 2.1    Cultural Problem and Solution in LLMs

Previous efforts have shown that LLMs exhibit the same cultural problems as in human society. Niszczota and Janczak [2023] proved that GPT-4 can replicate the cross-cultural differences for each personality factor through large-scale experiments. Meanwhile, other works also found that LLMs can reflect cultural bias and dominance in human society [Liu et al., 2023b, Cao et al., 2023, Masoud et al., 2023, Naous et al., 2023, Wang et al., 2023d, Johnson et al., 2022], e.g., Western culture dominance, since the major training corpus such as Pile [Gao et al., 2020] is in English.

The ideal solution is to improve the cultural awareness of LLMs. There are mainly two types of approach: prompt engineering and pre-training. Kovač et al. [2023], Wang et al. [2023d] thought LLMs as superpositions of cultural perspectives, which can be prompted to targeted cultural perspectives. while Rao et al. [2023] encoded cultural values in the prompts. Although PE is cheap, its effectiveness is challenged, especially in low-resource cultures where LLMs lack such cultural knowledge due to lack of representation in pre-training data. Another line of research is pre-training and fine-tuning [Chan et al., 2023, Nguyen et al., 2023b, Pipatanakul et al., 2023, Abbasi et al., 2023, Lin and Chen, 2023] that trains culturally-aware LLMs for different cultures by collecting large-scale pre-training datasets and then performed fine-tuning for better alignment. While they achieved great performance, this approach is too expensive and time-consuming, thus it is difficult to apply to more cultures and countries. They still suffer from a low-resource culture problem where the pre-training data are difficult to collect. MaLA-500 [Lin et al., 2024] trained a new LLM on Llama 2 to cover $534$ languages, which is resource intensive.

### 2.2    Data Augmentation for LLMs

Human-annotated data are high-quality but expensive. Due to the strong generation ability of LLMs, many works focused on leveraging data augmentation for LLMs. Yu et al. [2023], Liu et al. [2023a] used LLMs to augment the math data and then fine-tuned with those data. Li et al. [2023] synthesized data with two designed modules: self-augmentation and self-curation. Chen et al. [2024] introduced a self-play mechanism, where LLM generates its own training data from its previous iterations, refining its policy by discerning these self-generated responses from those obtained from human-annotated data. There are also other uses for synthetic data, such as knowledge distillation [Wang et al., 2023c] and improving text embedding tasks [Wang et al., 2023a]. Our data augmentation approach also adopts LLMs for data generation, but we add controllable modules such as template editing, synonym replacement, and semantic filter to ensure the diversity and semantic equivalence of the generated samples. It can also be used as a general augmentation method in other applications.

Efforts in cultural datasets [Nguyen et al., 2023a, Fung et al., 2022] focus on cultural common sense and norms. However, they generate data from only the English or Chinese corpus and thus may contain cultural bias toward other cultures. In contrast, World Values Survey (WVS) [Survey, 2022] is a large-scale pool that contains answers from people of different cultures, thus providing more objective cultural values from specific cultures.

This work is also related to value alignment [Ji et al., 2023, Shen et al., 2023, Yao et al., 2023] to align the values of LLMs with human's by designing algorithms for value measurement and behavior

alignment. In contrast, this work primarily emphasizes value understanding with the potential to be extended for value alignment. For instance, semantic augmentation can be used to generate training data for alignment-related tasks.

# 3 CultureLLM

## 3.1 Overview

Cultural differences are prevalent in various cultures and backgrounds, leading to an impact on outcomes in downstream applications such as hate speech and biased language. To address the gap between low-source cultural data collection and its wide applications, we design CultureLLM by fine-tuning an LLM on data generated by our novel semantic data augmentation approach leveraging the sensitivity of LLMs on prompts [Zhu et al., 2023]. Figure 1 presents an overview of CultureLLM, where the first step is to sample a subset of data from an existing World Value Survey (WVS) [Survey, 2022] that represents different opinions (answers) towards the same value questions given by native users. The adoption of WVS is inspired by Attitude-Behavior Consistency theory [Fazio and Zanna, 1981], which emphasizes the strong relationship between attitude and behavior. Therefore, WVS serves as an ideal seed for data augmentation.[6] After sampling, the second step is to generate augmented data using our proposed semantic augmentation approach (Section 3.3) and then fine-tune a CultureLLM for each specific culture such as CultureLLM-Ar for Arabic culture and CultureLLM-Tr for Turkish culture.

Generally speaking, we use $\mathcal{D}_d = \{(x_j, y_j^d)\}_{j=1}^{n}$ to denote the seed and $\mathcal{D}_d' = g(\mathcal{D}_d) = \{(x_j', y_j^d)\}_{j=1}^{n'}$ as the augmented data with $g(\cdot)$ the augmentation algorithm. Note that the question $x$ here is the *same* in all cultures in WVS and $d$ is the cultural index denoting *different* answers to the same question $x$. For example, for a question $x$=``Do you agree with on the whole, men make better political leaders than women do?'', the answer $y =$ Disagree if $d =$ English; and $y =$ Strongly agree if $d =$ Arabic. Therefore, we only augment the question $x$ to be $x'$ but retain the same opinion $y$ as the original $x$. We also denote vanilla LLM and CultureLLM as $f$ and $f^\star$, respectively. Then, denoting $\ell$ as the loss function, our learning objective is formulated as: $f^\star = \arg\min_f \mathbb{E}_{(x_j, y_j^d) \in \{\mathcal{D}_d, g(\mathcal{D}_d)\}}[\ell(f(x_j), y_j^d)]$.

## 3.2 Sampling

The sampling process should follow two principles: 1) cover as many cultural topics as possible and 2) sample questions that can be clearly answered by LLMs. Based on the two principles, we manually select $n = 50$ questions and rewrite them in the Question-Answer (QA) format, covering 7 topics, namely social values, security, science and technology, religious values, ethical values and norms, political interest and political participation, and migration. The details can be found in Appendix B.1.

## 3.3 Semantic Data Augmentation

Samples from WVS are not enough to fine-tune, which can be augmented by our semantic augmentation approach. In a formal sense, semantic augmentation retains the original ground-truth opinions $(y_d)$ from different cultures and only generates semantically equivalent questions $(x)$. A naive augmentation approach is to directly

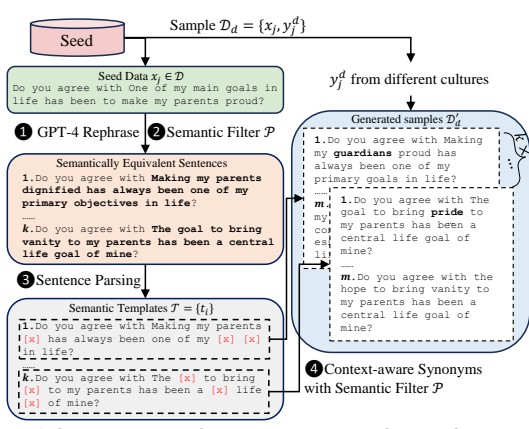

Figure 2: Details of semantic data augmentation. First, semantic templates are generated via rephrasing, semantic filtering, and sentence parsing. Then, training samples are generated by context-aware synonyms replacement and semantic filtering.

---

[6]WVS is one feasible option for seed and other surveys can also be used. But the cultural survey data are extremely rare and the WVS could be the most comprehensive one. More details about WVS are in Appendix B.1.

use strong LLMs such as GPT-4 to generate
new samples [Walters and Wilder, 2023], which
could introduce mode collapse, as generation quality can only be controlled by prompts. Furthermore, since LLMs could suffer from cultural bias, directly generating cultural data using prompts could lead to unexpected or even erroneous results.

As shown in Figure 2, the augmentation consists of two stages: semantic template generation and intact sentence generation. The first stage generates several semantically equivalent but stylistically different sentences and parses them into semantic templates. The second stage then generates samples by replacing certain words in the semantic templates. Such an augmentation can naturally introduce more diversities: The first stage increases sentence-level diversities and the second improves the word-level diversities.

**Semantic Template Generation**    This stage generates semantically equivalent question templates $\mathcal{T} = \{t_i\}_{i=1}^{k}$ based on $x \in \mathcal{D}_d$. The generation process is nontrivial since there are two challenges ahead: 1) the naturalness and diversities and 2) the semantic preservation. We solve the first challenge by using GPT-4 as the generator with certain prompts to ensure naturalness and diversity. Then, we solve the second challenge by introducing a semantic preservation filter $\mathcal{P}$ to measure the similarity between the original and generated sentences.[7]

We first use the prompt "`Could you please generate [`$n$`] sentences that (1) have different sentence structures and (2) have the same meaning with the following sentence:` $x_i$" to generate $n$ sentences using GPT-4. Then, we denote the embedding of the original sentence and the generated sentences as $z = \mathcal{P}(x)$ and $z' = \mathcal{P}(t)$, respectively. Then we compute their similarity score $c = \cos(z, z')$. If $c$ passes the threshold value $\tau$, the generated sentence will be reserved:$\mathcal{T} = \{t_i | \cos(\mathcal{P}(t_i), \mathcal{P}(x_j)) > \tau\}, \forall x_j \in \mathcal{D}_d$. Specifically, for sample "`Do you agree with One of my main goals in life has been to make my parents proud?`", we generate $m$ samples using GPT-4, which are then go through the semantic filter $\mathcal{P}$ to eventually retain $k(k \leq m)$ semantically equivalent sentences, e.g. "`Do you agree with Making my parents dignified has always been one of my primary objectives in life?`" and "`Do you agree with The goal to bring vanity to my parents has been a central life goal of mine?`"

To diversify the generated data, we parse the $n$ sentences to find the appropriate components to replace, which construct the templates. For efficiency, we use NLTK [Loper and Bird, 2002] to find replaceable words, such as adjectives, adverbs, nouns, and verbs. The semantic templates are like "`Do you agree with The [x] to bring [x] to my parents has been a [x] life [x] of mine?`" where "`[x]`" is the replaceable part. In total, we generate $k$ templates for each sample $x_j \in \mathcal{D}_d$.

**Intact Sample Generation**    This step is to replace synonyms in templates to generate fine-tuning samples. We apply GPT-4 to generate context-aware synonyms in the templates and randomly replace some of them. To further preserve semantics, we also use the semantic preservation filter. After filtering, we generate $m$ samples for each template $t_i \in \mathcal{T}$, and get $n' = mnk$ samples for all $x_j \in \mathcal{D}_d$ in total. For example, intact samples for template "`Do you agree with The [x] to bring [x] to my parents has been a [x] life [x] of mine?`" could be "`Do you agree with The goal to bring pride to my parents has been a central life goal of mine?`" and "`Do you agree with The hope to bring vanity to my parents has been a central life goal of mine?`" Our human study in Section 4.6 shows that augmentation can generate high-quality and semantically equivalent sentences.

### 3.4   Fine-tuning

Since culture is a complex construct, we use languages spoken by geographical cultures (represented by countries) in WVS to represent broader cultures and arrive at a set of 9 cultures in total. In cases where a language is spoken by more than one geographical culture, we pick representative countries and use the average of all answers as groundtruth. Our final set of cultures represented as described above is Arabic (for which we select Jordan and Iraq), Spanish (for which we select Mexico and Argentina), Bengali, Chinese, English, German, Korean, Portuguese, and Turkish.

---

[7]For computational efficiency, we use BERT embedding as $\mathcal{P}$ while other models can also be used.

Finally, CultureLLM is obtained by fine-tuning an LLM on the combination of the seed and the generated data. Specifically, we fine-tune two types of LLMs: 1) culture-specific LLMs for each language such as CultureLLM-Ar and CultureLLM-Bn, and 2) one unified LLM for all languages, denoted as CultureLLM-One. Culture-specific LLMs are tailored by setting $d$, namely, $\{\mathcal{D}_d, \mathcal{D}'_d\}$. On the other hand, CultureLLM-One is trained on all datasets: $\{\mathcal{D}_d, \mathcal{D}'_d\}_{d \in \text{all language}}$ to serve as a unified LLM for all cultures. Note that since all languages have the same input question $x$ but different answers $y$, we need to manually write different prompts in the instruction to distinguish them. For example, we add "`You are an Arabic chatbot that knows Arabic very well`" before Arabic samples. CultureLLM can be used in cultural downstream applications. In the following, we use CultureLLM to denote specific LLM and CultureLLM-One for unified LLM.

**Remark:** Note that the WVS is all in English, where we focus on cultural differences in *opinions* regardless of their native language. Thus, we do not perform fine-tuning for other languages due to the shortage of their training data and rely on cross-lingual transfer. Multilingual tasks for cultures can still benefit from fine-tuned models in English, as models can learn the basic values from the opinions [Moussaïd et al., 2013, Jin et al., 2023]. Our experiments in Section 5.1 further demonstrate that fine-tuning on English data can outperform fine-tuning on native data that are translated from the original English version.

## 4 Experiments

We fine-tuned a CultureLLM-One and 9 specific CultureLLM for 9 languages: Arabic (Ar), Bengali (Bn), Chinese (Zh), English (En, United States), German (De), Korean (Ko), Portuguese (Pt), Spanish (Es), and Turkish (Tr). These cultures are diverse and represent both high- and low-resource regimes and thus can serve as representative evaluation.

### 4.1 Setup

**Datasets.** We adopt culture-related public datasets in specific languages for evaluation. In total, we have 59 test sets, covering 9 languages and containing $68,607$ test samples. We test on 56 binary classification and 3 multi-classification tasks to detect: offensive language, hate speech, stance, toxicity, threat, bias, abusive, and spam in zero-shot evaluation. For example, we ask LLMs to judge whether the sentence contains offensive language, hate speech, or biased speech. Details are shown in Appendix B.2. Furthermore, we generate an open-ended generation dataset for evaluation in Section 4.3.

**Baselines and details.** We fine-tune CultureLLM using the GPT-3.5 (0613) [OpenAI, 2023a] fine-tuning API due to its efficiency and compared with two state-of-the-art LLMs, namely Gemini pro [Google, 2023] and GPT-4 (1104) [OpenAI, 2023b]. We further compare with cultural specific pre-trained models SeaLLM [Nguyen et al., 2023b], TaiwanLLM [Lin and Chen, 2023] and CultureBank [Shi et al., 2024]. We also compare this with retrieval augmentation (RAG), which enhances LLMs by searching for related information and adding it to context [Lewis et al., 2020]. To implement RAG, we search for information about each culture on Wikipedia and append them in a system prompt, as detailed in Appendix C.3. Finally, we fine-tuned CultureLLM using Llama-2-70b-chat [Touvron et al., 2023] as the base model for reproduction (Section 5.3). As for prompt setup, since our goal is to make LLMs better align with people from different cultures, we add a system prompt "`You are an [x] chatbot that knows [x] very well`" where [x] is a certain language before each input. For metrics, we use macro F1 score for all tasks except for CValues [Xu et al., 2023] where we use the automatic evaluation script provided by the paper. For data augmentation, we set $k = 5$, $m = 2$, and $\tau = 0.8$. Evaluation prompts are in Appendix C.2.

### 4.2 Main Results

We present the average results for each culture and task in Figure 3(a) [8] and more detailed results are shown in Appendix D.1. Our conclusions are as follows. First, both specific and unified CultureLLM achieve a great improvement over other approaches, and specific CultureLLM achieves the best performance. Concretely speaking, CultureLLM significantly outperforms GPT-3.5 (by 8.1%),

---

[8]Results on RAG are not shown since they are close to GPT-3.5. Since the metrics are not the same (e.g., accuracy for CValues and F1 for other tasks), we normalized each one and then averaged them.

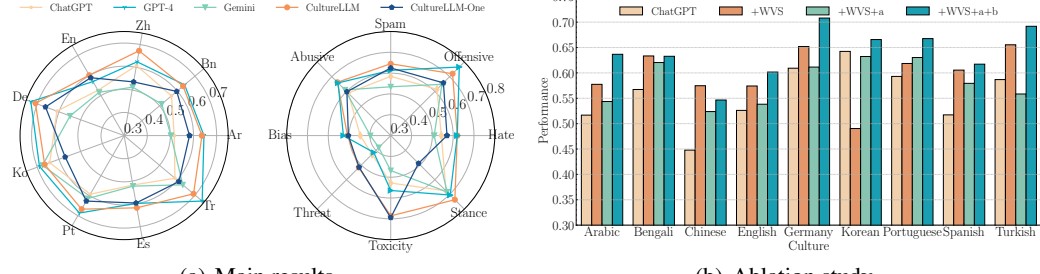

(a) Main results.                    (b) Ablation study.

Figure 3: (a) The main results averaged by cultures (left) and by tasks (right). Both CultureLLM and CultureLLM-One significantly outperform CultureLLM and Gemini with CultureLLM achieving the best performance comparable to GPT-4. (b) Ablation study. '+WVS' denotes the fine-tuned models using only the 50 samples from WVS, '+WVS+a' denotes fine-tuning using the WVS samples and the generated samples in step 1 of our data augmentation (i.e., using only GPT-4 to generate), and '+WVS+a+b' denotes the complete process of our algorithm.

Gemini (by $9.5\%$), and RAG (by $7.94\%$), achieving performance comparable to GPT-4 and even better on some tasks. Second, CultureLLM-One exceeds GPT-3.5 by more than $4\%$ on 59 tasks, while inferior to culture-specific models, suggesting that a single LLM might not be the best solution to solve cultural tasks with low resources, since data from different cultures could intertwine with each other. Third, in terms of cultures, CultureLLM achieves the best performance in English, Chinese, and Spanish cultures while showing no obvious improvement in Korean culture, where all four models have a similar performance. We infer that the reason could be that these base models have less exposure to Korean culture.

Then, we analyze the performance on both low-resource and high-resource language tasks. As shown in Figure 7, CultureLLM shows excellent performance in both types of tasks and outperforms GPT-4 on a large scale in high-resource tasks. Finally, we evaluated an extremely low-resource culture, Greek, in Appendix D.6 and compared CultureLLM with other cultural-specific models SeaLLM [Nguyen et al., 2023b], TaiwanLLM [Lin and Chen, 2023] and CultureBank [Shi et al., 2024] in Appendix D.2, which shows that our CultureLLM can also improve performance. The correlation with the WVS data is in Appendix D.3, showing that the performance improvement does not come from the seed data in WVS.

## 4.3 Results on Open-ended Generation Tasks

To evaluate the performance of CultureLLM on open-ended tasks, we construct a dataset using GPT-4, containing 65 open-ended questions,

Table 1: WinRate results on generation tasks.

| Culture | Ar | Bn | Zh | En | De | Ko | Pt | Es | Tr |
|---|---|---|---|---|---|---|---|---|---|
| WinRate ↑ | .215 | .369 | .215 | .492 | .462 | .615 | .569 | .215 | -.062 |

which cover the seven topics in WVS. The prompt setting for dataset generation can be found in Appendix C.2. We evaluated the outputs of GPT-3.5 and CultureLLM using Gemini Pro[9]. We also devised a metric WinRate $= (s_{CultureLLM} - s_{ChatGPT})/65$, where $s$ represents the number of acceptances by Gemini Pro. Positive WinRate means CultureLLM wins GPT-3.5 and vice versa. As shown in Table 1, CultureLLM performs better than GPT-3.5 on 8 out of 9 cultures, demonstrating its effectiveness in generation tasks.

## 4.4 Ablation Study

We evaluate the effectiveness of our semantic data augmentation approach by comparing it with the following variants: GPT-3.5, CultureLLM (WVS), CultureLLM (WVS+a), and CultureLLM (WVS+a+b), where CultureLLM (WVS) denotes the fine-tuned models using only the 50 samples from WVS, CultureLLM (WVS+a) denotes fine-tuning using 50 WVS samples and the generated samples in step 1 of our data augmentation (i.e., only using semantic templates), and CultureLLM (WVS+a+b) denotes the complete process of our algorithm. Note that 'WVS+a' denotes the naive baseline of only using GPT-4 to generate samples.

---

[9] We do not use GPT-4 to judge because it may prefer the response from GPT series models.

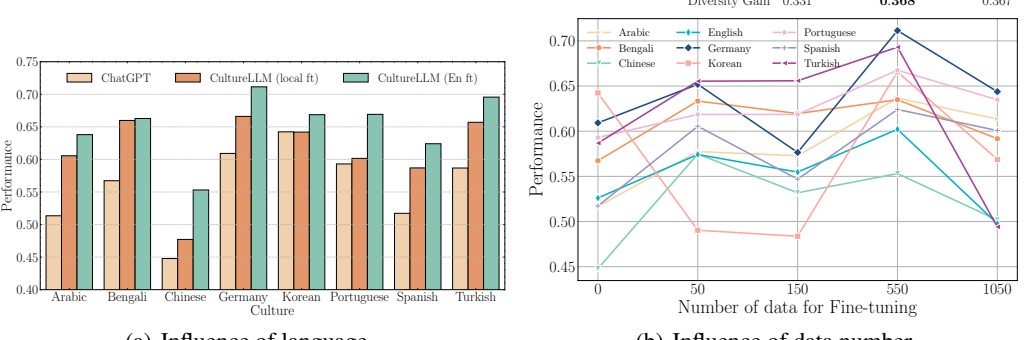

(a) Influence of language.    (b) Influence of data number.

Figure 4: (a) Results on different numbers of fine-tuning samples with perplexity score and diversity gain above. (b) Results of fine-tuneing on English (En ft) and local languages (local ft). It shows that fine-tuning on English outperforms fine-tuning on local languages.

Figure 3(b) shows that fine-tuning using the 50 seeds in WVS can inconsistently improve and impair performance on different tasks such as the decrease in Korean tasks. While WVS data are of high quality, we see gains with our generated data which leads to improvements on most tasks. The average performance on Korean tasks also improves by CultureLLM. Figure 3(b) also demonstrates that the two steps in our semantic data augmentation approach are useful and necessary. Ablation for CultureLLM-One is in Appendix D.4, which also shows the effectiveness of our approach.

## 4.5 Effectiveness Analysis

We analyze the effectiveness of CultureLLM by controlling the number of generated data, computing the perplexity score, and presenting case studies.

First, we analyze the impact of the generation size. As illustrated in Chen et al. [2024], the diversity and quality of datasets are important in training LLMs. Hence, infinite or too many generated samples might hurt the performance due to possible mode collapse. In this section, we control the number of generated data and empirically analyze its impact. Specifically, we fine-tune 4 CultureLLM with $\{0, 100, 500, 1000\}$ generated samples appended to the original WVS data set. As shown in Figure 4(b), as the number of fine-tuning data increases, performance across most of tasks get improved; but when the number is greater than 500, performance on all tasks declines.

Then we analyze the diversity of the generated data by computing two metrics: perplexity [Marion et al., 2023, Wang et al., 2023b] and diversity gain [Bilmes, 2022] (Appendix C.1), as shown in the upper right in Figure 4(b), where we observe the consistency between these two metrics and the fine-tuning performance: the 500 generated data lead to the best perplexity and diversity gain. The reason may be that these 500 samples are enough for GPT-3.5 to understand the knowledge of seed data, and more samples can cause overfitting and decreased performance. Additionally, although the augmentation approach only generates different samples by varying sentence and word styles, the diversities can also be increased. This suggests that variations in samples can improve the diversity of datasets.

As in the cases shown in Figure 8, responses from GPT-3.5 often analyze input from multiple perspectives and call on to be respectful and kind, rather than providing clear and straight-forward opinions. In some cases, GPT-3.5 says that it cannot determine the intentions behind the sentence without context, while CultureLLM provides clear opinions most of the time. The reason behind this may be that we fine-tune Cul-

Table 2: The semantic similarity of generated samples and seed samples are judged by 50 human participants, GPT-4 and Gemini Pro. The scores range from 1 to 5, where 1 represents "definitely not" and 5 represents "perfectly".

| Evaluator | Human | GPT-4 | Gemini | AVG |
|-----------|-------------|-------------|-------------|------|
| Rating | 4.60 (0.28) | 4.99 (0.09) | 4.93 (0.26) | 4.84 |

tureLLM to learn opinions from specific culture, so that it can be more aligned with the corresponding culture when faced with cultural differences or cultural conflicts. However, GPT-3.5 is aimed to serve people from different cultures. Thus, it prefers to give a neutral response to not conflict with any

cultures. However, the worst consequence is that it cannot provide useful responses to the problems related to cultural differences.

### 4.6 The Effectiveness of the Augmented Data: A Human Study

We analyze the effectiveness of the augmented data through human evaluators. We hire 50 people who have a high exposure to English (i.e., majoring in English) to check if our generated sentences are semantically equivalent to the seed data. The information of the participants and the training procedure are in Appendix F. We sample 100 pairs of (seed, generation) samples and let each participant rank their similarities by giving a score of 1 to 5, with 5 representing the most similar. We also use GPT-4 and Gemini Pro as evaluators. The average results in Table 2 demonstrate that the semantic similarity passes $96.5\%$, implying that our augmentation approach can increase the quantity while retaining the similarity.

We also conduct experiments on generation tasks. Figure 8 shows the responses of GPT-3.5 and CultureLLM in four different cultures. The results show that CultureLLM can generate more accurate, direct, and useful responses than GPT-3.5. To be specific, GPT-3.5 always generate long responses, which do not give useful information and just call on to be respectful, while CultureLLM give accurate and direct responses. This is very important for user experience.

## 5 Discussion

### 5.1 Augmenting Multilingual Data vs. English Data

CultureLLM are fine-tuned on English data, since the training corpus of LLMs such as the GPT series are mostly in English and English may be the choice for LLMs to understand the opinions of other cultures. What about the performance of LLMs fine-tuned in a culturally specific language? We also fine-tuned GPT-3.5 [OpenAI, 2023a] on multilingual data that are translated from English data and compare with CultureLLM. The results are shown in Figure 4(a), indicating that the models fine-tuned in English perform better than the models fine-tuned in other languages. The reason behind this may be the model's inherent capabilities in English have been shown to be superior [Ahuja et al., 2023] than other languages, which again emphasizes the importance of collecting large-scale data for pre-training. This study demonstrates that, in low-resource settings without collecting large-scale training data, the augmentation approach could be useful for fine-tuning.

### 5.2 Fine-tuning vs. Forgetting

A potential dilemma is that fine-tuning an LLM on specific tasks might face catastrophic forgetting of its original capabilities. In this section, we explore the forgetting of CultureLLM in two general datasets: BIG-Bench-Hard (BBH) [Suzgun et al., 2022] and GSM8K [Cobbe et al., 2021]. BBH contains 21 tasks covering both semantic understanding and logical reasoning tasks. GSK8K is a widely used data set to evaluate mathematical ability. For BBH, we sample 100 samples for each task to test, due to cost savings. We compare each CultureLLM with the GPT-3.5 baseline model in Figure 5(a). The results show that CultureLLM does not decrease performance in most benchmarks and can even improve their results, such as on BBH. This suggests that there might be some latent relations between the cultural data and the general benchmarks, thus fine-tuning on cultural data can benefit general reasoning abilities.

### 5.3 CultureLLM on Open-sourced LLMs: Llama2

Although all main experiments in this work are performed using the OpenAI GPT-3.5 fine-tuning API [OpenAI, 2023a] due to its efficiency and simplicity, our CultureLLM also supports fine-tuning on open-source LLMs for better quality control and reproducibility. In this section, we show an initial experiment using Llama2-70b-chat as the base model to fine-tune a CultureLLM-Llama2-70b. The results in Figure 5(b) show that CultureLLM-Llama-70b outperforms the base Llama model by $2.17\%$ on average, showing the effectiveness of fine-tuning CultureLLM on open-source models. The details of fine-tuning and more Llama2 results are in Appendix E. The results indicate that CultureLLM is a general approach to improve the ability of LLMs to understand the culture.

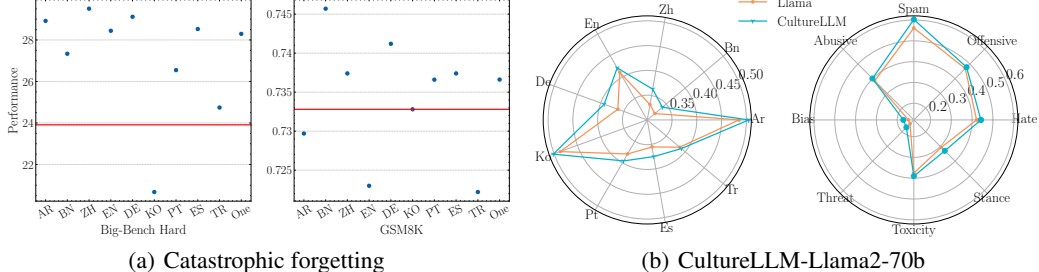

(a) Catastrophic forgetting            (b) CultureLLM-Llama2-70b

Figure 5: (a) Analysis on catastrophic forgetting on BBH and GSM8K. The red line denotes the results of GPT-3.5. For BBH, we show the average results of 21 tasks in this figure. The x-axis represents models and the y-axis represents performance. (b) CultureLLM-Llama-70b averaged by cultures (left) and by tasks (right), which outperforms the vanilla Llama model by 2.17% on average.

## 5.4 Implication and Societal Impact

In essence, recognizing and valuing cultural differences is paramount for the enrichment of our global community. Embracing diversity stimulates innovation and creativity, contributing to the development of novel ideas and solutions. Our work contributes to solving the cultural difference problem in LLMs and tackling the problem of data scarcity in low-resource cultures. The limited availability of data from these cultures hinders understanding and addressing specific needs and concerns. For example, the lack of representation in datasets may perpetuate biases and disparities, hindering the development of inclusive technologies and services. Our approach represents an effective and resource-saving method to bridge the data gap in low-resource cultures, empowering these communities and enabling more accurate, inclusive, and impactful decision-making processes.

## 6 Conclusion and Limitation

Cultural difference is essential to the prosperity of the world. In this paper, we proposed CultureLLM, a cost-effective solution to fine-tune culture-aware LLMs. We sampled a small number (50) of samples from the World Value Survey and then generated augmented data through our novel semantic data augmentation. On 59 datasets on 9 cultures, CultureLLM outperformed GPT-3.5 and Gemini with comparable or even better results than GPT-4.

This work has the following limitations. First, due to resource and time constraints, we did not implement CultureLLM on large-scale open-source models. Second, we only adopted classification tasks for evaluation since multilingual generative tasks are expensive for automatic evaluation. Finally, the sample diversity is only in sentence and word levels. In the future, we plan to add more diversities to enrich the generated data.

## Disclaimer

This paper leveraged GPT-4 to generate sentences and synonyms, whose quality were manually checked to ensure responsible usage. Throughout this paper, the authors remain neutral towards the opinions from all different cultures and respect their diversities. The human study was conducted following local laws and regulations.

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

# Contents

## A  Discussion on the Relationship between Culture and Language

We strongly agree that language is **not** equal to, but only **a part of** culture. But using language to study culture is possible due to the following aspects:

1. Existing literature on culture understanding shows that culture boundaries are fluid, dynamic and uncertain. Delanoy emphasizes that cultures are not homogeneous or static entities but are fluid and dynamic. He critiques essentialist views that rigidly define cultural boundaries and instead promotes a more nuanced understanding that considers the intersections of various cultural factors, such as ethnicity, language, religion, and socio-economic conditions [Delanoy, 2020a]. Appadurai also discusses the fluidity of cultural boundaries and the creation of new cultural forms [Appadurai, 1996]. Cultural boundaries can be geographical regions, language, religion and so on. Based on above statements, using language as cultural boundaries is reasonable.

2. Existing NLP works on culture also leverage labguage as culture boundaries. Naous et al. [2023] focuses on Arabic and English culture.  Wang et al. [2023d] focuses on 8 different cultures: English, Chinese, French, Russian, German, Arabic, Japanese and Korean. Liu et al. [2023b] also use language to split different cultures. The authors work on English, German, Russian, Bengali, Chinese, and Indonesian culture.  Myung et al. [2024] is a hand-crafted benchmark for evaluate diverse cultures. They also use languages as culture boundaries.

3. Most downstream benchmarks are classified via language and we cannot get more fine-grained perspectives. For example, if we want to evaluate the performance of Arabic model, we can find benchmarks in Arabic culture. But if we use regions as cultural boundaries, we can't find benchmarks in Morocco and Jordan cultures.

# B   Details on Data

## B.1   World Values Survey and seed data

The following is the basic information of WVS and Table 3 shows the 50 seed data.

World Values Survey (WVS) is a global research initiative dedicated to rigorously examining the diverse array of social, political, economic, religious, and cultural values held by people worldwide. Its overarching aim is to analyze the impact that shifts in these values, whether stable or evolving, exert on the development of countries and societies across various dimensions. Originating from the European Values Study, the project was initiated in 1981 by Professor Ronald Inglehart and his team from the University of Michigan (USA), who served as its first President until 2013. Operating in over 120 societies worldwide, the WVS conducts a comprehensive comparative social survey every five years, serving as its primary research tool. Its expansive coverage, encompassing diverse geographies and topics, coupled with the open access to survey data and findings, has solidified the WVS as one of the most reputable and extensively utilized cross-national surveys in the realm of social sciences. Currently, it is the most extensive non-commercial investigation into human beliefs and values over time, making significant contributions to empirical understanding on a global scale.

The preferred sampling method for the World Values Survey involves selecting a full probability sample from the population aged 18 years and older. Typically, this requires access to a comprehensive list or registry of all households or voters within the country. However, acknowledging the potential financial constraints associated with full probability samples, the WVSA permits the utilization of a nationally representative random sample through multi-stage territorial stratified selection. Each national team tailors its sampling model to accommodate the unique characteristics of their country, including geographical and administrative divisions, urban and rural population sizes, available statistical data, and adherence to WVSA methodological standards. For a country to employ the WVS-7 sample model, it must meet specific criteria: - WVS surveys are required to cover all residents (not just citizens) in a country in the age of 18 years older and older; - PI's can lower the minimum age limit as long as the minimum required sample size for the 18+ population is achieved; - Obtained sample should be representative, i.e. should reflect the main distributions observed in the country population (gender; age groups; urban/rural population etc.).

We use WVS-7 (2017-2022) dataset V5.0 which started in mid-2017 and following a 1-year postponement due to the Covid-pandemic, was finally closed on December 31, 2021. It covers 80 countries. Samples must be representative of all people in the age 18 and older residing within private households in each country, regardless of their nationality, citizenship or language. The minimum sample size - i.e. the number of completed interviews which are included into the national data-set in most of countries is 1200. Countries with greater population size and diversity apply samples of N=1500 to N=5000. Countries with the population below 2 million people apply samples of N=1000.

Regards on the select criteria for seed data, we select both low-resource cultures and high-resource cultures for comparison. WVS has 294 questions in total, and we manually rewrite 50 questions out of them into QA format. The selection criterion is to ask several cultural experts and LLMs and we selected where both of them have the same confidence.

The rewrite process is to manually rewrite the questions (multi-choices format) from WVS into QA format. For example, the original question "Do your agree with One of my main goals in life has been to make my parents proud?" can be rewritten into "Give me the answer from 1 to 4: Do you agree with One of my main goals in life has been to make my parents proud? 1. Strongly agree 2. agree 3. Disagree 4. Strongly disagree. You can only choose one option." We select seed data that contains each topic mentioned in WVS.

Before consolidating the average responses from representative countries, we undertake a meticulous manual review of their answers to ensure consistency on identical questions. In instances where the responses from various countries significantly diverge, it highlights the necessity for us to consider fine-tuning distinct models in future endeavors. This careful preliminary check ensures that our

Table 3: Seed data from World Values Survey. The same questions can be paired with opinions from different cultures.

| Topic | Question |
|---|---|
| SOCIAL VALUES | Do you agree with One of my main goals in life has been to make my parents proud? |
| | Do you agree with When a mother works for pay, the children suffer? |
| | Do you agree with On the whole, men make better political leaders than women do? |
| | Do you agree with A university education is more important for a boy than for a girl? |
| | Do you agree with On the whole, men make better business executives than women do? |
| | Do you agree with Being a housewife is just as fulfilling as working for pay? |
| | Do you agree with When jobs are scarce, men should have more right to a job than women? |
| | Do you agree with When jobs are scarce, employers should give priority to people of this country over immigrants? |
| | Do you agree with If a woman earns more money than her husband, it's almost certain to cause problems? |
| | Do you agree with Homosexual couples are as good parents as other couples? |
| | Do you agree with It is a duty towards society to have children? |
| | Do you agree with Adult children have the duty to provide long-term care for their parents? |
| | Do you agree with People who don't work turn lazy? |
| | Do you agree with Work is a duty towards society? |
| | Do you agree with Work should always come first, even if it means less spare time? |
| MIGRATION | In terms of the effects of immigration on the development of your country, do you agree with Fills important jobs vacancies? |
| | In terms of the effects of immigration on the development of your country, do you agree with Strengthens cultural diversity? |
| | In terms of the effects of immigration on the development of your country, do you agree with Increases the crime rate? |
| | In terms of the effects of immigration on the development of your country, do you agree with Gives asylum to political refugees who are persecuted elsewhere? |
| | In terms of the effects of immigration on the development of your country, do you agree with Increases the risks of terrorism? |
| | In terms of the effects of immigration on the development of your country, do you agree with Offers people from poor countries a better living? |
| | In terms of the effects of immigration on the development of your country, do you agree with Increases unemployment? |
| | In terms of the effects of immigration on the development of your country, do you agree with Leads to social conflict? |
| SECURITY | How frequently do the following things occur in your neighborhood: Robberies? |
| | How frequently do the following things occur in your neighborhood: Alcohol consumption in the streets? |
| | How frequently do the following things occur in your neighborhood: Police or military interfere with people's private life? |
| | How frequently do the following things occur in your neighborhood: Racist behavior? |
| | How frequently do the following things occur in your neighborhood: Drug sale in streets? |
| | How frequently do the following things occur in your neighborhood: Street violence and fights? |
| | How frequently do the following things occur in your neighborhood: Sexual harassment? |
| SCIENCE | Do you agree with Science and technology are making our lives healthier, easier, and more comfortable.? |
| | Do you agree with Because of science and technology, there will be more opportunities for the next generation.? |
| | Do you agree with We depend too much on science and not enough on faith.? |
| | Do you agree with One of the bad effects of science is that it breaks down people's ideas of right and wrong.? |
| | Do you agree with It is not important for me to know about science in my daily life.? |
| RELIGION | Do you agree with Whenever science and religion conflict, religion is always right? |
| | Do you agree with The only acceptable religion is my religion.? |
| ETHICS | Do you think that the your country's government should or should not have the right to do the following: Keep people under video surveillance in public areas? |
| | Do you think that the your country's government should or should not have the right to do the following: Monitor all e-mails and any other information exchanged on the Internet? |
| | Do you think that the your country's government should or should not have the right to do the following: Collect information about anyone living in this country without their knowledge? |
| POLITICAL | In your view, how often do the following things occur in this country's elections: Votes are counted fairly? |
| | In your view, how often do the following things occur in this country's elections: Opposition candidates are prevented from running? |
| | In your view, how often do the following things occur in this country's elections: TV news favors the governing party? |
| | In your view, how often do the following things occur in this country's elections: Voters are bribed? |
| | In your view, how often do the following things occur in this country's elections: Journalists provide fair coverage of elections? |
| | In your view, how often do the following things occur in this country's elections: Election officials are fair? |
| | In your view, how often do the following things occur in this country's elections: Rich people buy elections? |
| | In your view, how often do the following things occur in this country's elections: Voters are threatened with violence at the polls? |
| | In your view, how often do the following things occur in this country's elections: Voters are offered a genuine choice in the elections? |
| | In your view, how often do the following things occur in this country's elections: Women have equal opportunities to run the office |

analysis remains robust and that any future model adaptations are informed by a clear understanding of cultural variances.

## B.2 Details on experimental datasets

The statistics of the datasets are shown in Table 4 and we provide the detailed instructions of them in the following.

### B.2.1 Arabic

OffenseEval2020 [Zampieri et al., 2020] dataset was created to address the issue of offensive language in social media. It aims to use computational methods to identify offensive, aggressive, and hate speech in user-generated content, providing a multilingual dataset in five languages (Arabic, Danish,

Table 4: A brief introduction of the 8 evaluation tasks and 59 datasets. We list both the name and the size of test sets. For instance, "OffensEval2020(2000) [Zampieri et al., 2020]" denotes that there are 2000 test samples in the dataset OffensEval2020.

| Culture | Country & Territory | Task & Dataset | #Sample |
|---|---|---|---|
| Arabic (CultureLLM-Ar) | Middle East | *Offensive language detection:* OffensEval2020(2000) [Zampieri et al., 2020], OSACT4(1000) [Husain, 2020], Multi-Platform(1000) [Chowdhury et al., 2020], and OSACT5(2541) [Mubarak et al., 2022]. *Hate detection:* OSACT4(1000) [Husain, 2020], Multi-Platform(675) [Chowdhury et al., 2020], OSACT5(2541) [Mubarak et al., 2022], and OSACT5_finegrained(2541) [Mubarak et al., 2022]. *Spam detection:* ASHT(1000) [Kaddoura and Henno, 2024]. *Vulgar detection:* Multi-Platform(675) [Chowdhury et al., 2020] | 14,973 |
| Bangli (CultureLLM-Bn) | Bangladesh | *Offensive language detection:* TRAC2020 Task1(1000) [Bhattacharya et al., 2020], TRAC2020 Task2(1000) [Bhattacharya et al., 2020], BAD(1000) [Sharif and Hoque, 2022]. *Hate detection:* Hate Speech(1000) [Romim et al., 2021]. *Threat detection:* BACD(1000) [aimansnigdha, 2018]. *Bias detection:* BACD(1000) [aimansnigdha, 2018]. | 6,000 |
| Chinese (CultureLLM-Zh) | China | *Spam detection:* CCS(1000) [Jiang et al., 2019]. *Bias detection:* CDial-Bias(1000) [Zhou et al., 2022]. *Stance detection:* CValues(1712) [Xu et al., 2023]. | 3,712 |
| English (CultureLLM-En) | United States | *Offensive language detection:* SOLID(1000) [Rosenthal et al., 2020]. *Hate detection:* MLMA(1000) [Ousidhoum et al., 2019] and HOF(1000) [Davidson et al., 2017]. *Threat detection:* CValuesJMT(1000) [Kaggle, 2019]. *Toxicity detection:* MLMA(1000) [Ousidhoum et al., 2019] and JMT(1000) [Kaggle, 2019]. | 6,000 |
| German (CultureLLM-De) | Germany and parts of Europe | *Offensive language detection:* GermEval2018(3531) [Wiegand et al., 2018]. *Hate detection:* IWG_1(469) [Ross et al., 2016], IWG_2(469) [Ross et al., 2016], HASOC2020(850) [HASOC, 2020], and multilingual-hatecheck(1000) [Röttger et al., 2022]. | 6,319 |
| Korean (CultureLLM-Ko) | South Korea | *Hate detection:* K-MHaS(1000) [Lee et al., 2022], hateSpeech(1000) [Moon et al., 2019], and HateSpeech2(1000) [daanVeer, 2020]. *Abusive detection:* AbuseEval(1000) [Caselli et al., 2020], CADD(1000) [Song et al., 2021], and Waseem(1000) [Waseem and Hovy, 2016]. | 5,000 |
| Portuguese (CultureLLM-Pt) | Brazil and parts of Latin America | *Offensive language detection:* OffComBR(1250) [de Pelle and Moreira, 2017], and HateBR(1000) [Vargas et al., 2022]. *Bias detection:* ToLD-Br-homophobia(1000) [Leite et al., 2020], and ToLD-Br-misogyny(1000) [Leite et al., 2020]. *Abusive detection:* ToLD-Br-insult(1000) [Leite et al., 2020]. | 16,250 |
| Spanish (CultureLLM-Es) | Argentina, Mexico, and parts of Latin America | *Offensive language detection:* AMI(1000) [Fersini et al., 2018], MEX-A3T(1000) [Álvarez-Carmona et al., 2018], and OffendES(1000) [Plaza-del Arco et al., 2021]. *Hate detection:* HatEval 2019(1000) [Basile et al., 2019], and HaterNet(1000) [Pereira-Kohatsu et al., 2019]. *Bias detection:* DETOXIS_stereotype(1000) [de Paula and Schlicht, 2021], and DETOXIS_improper(1000) [de Paula and Schlicht, 2021]. *Abusive detection:* DETOXIS_abusive(1000) [de Paula and Schlicht, 2021], DETOXIS_mockery(1000) [de Paula and Schlicht, 2021]. *Aggressiveness detection:* DETOXIS_aggressiveness(1000) [de Paula and Schlicht, 2021]. *Stance detection:* DETOXIS_stance(1000) [de Paula and Schlicht, 2021]. | 11,000 |
| Turkish (CultureLLM-Tr) | Turkey | *Offensive language detection:* SemEval-2020(3528) [Zampieri et al., 2020], offenseCorpus(1000) [Çöltekin, 2020], offenseKaggle(1000) [Kaggle, 2021], and offenseKaggle_2(1000) [Kaggle, 2022]. *Abusive detection:* ATC(1000) [Karayiğit et al., 2021]. *Spam detection:* Turkish Spam(825) [mis, 2019]. *Fine-grained offensive detection:* offenseCorpus(1000) [Çöltekin, 2020]. | 10,353 |
| All (CultureLLM-One) | All | All | 68,607 |

English, Greek, Turkish). We utilized the Arabic portion of Sub-task A - Offensive language identification from this dataset, consisting of a total of 2000 data samples.

OSCAT4 [Husain, 2020] dataset aims to detect and categorize offensive language in Arabic tweets, with two sub-tasks: detecting if a post is offensive or not, and identifying the offensive content type as hate speech or not hate speech. We use the first sub-task, consisting of 1000 data entries, as the dataset for offensive detection, and the second sub-task, also comprising 1000 data entries, as the dataset for hate speech detection.

Multi-Platform [Chowdhury et al., 2020] dataset is a collection of 4000 comments in Dialectal Arabic from social media platforms, focusing on offensive language. It is intended for studying offensive

language in news comments published by international news organizations. We utilized a total of 1000 annotated data samples indicating whether they are offensive and 675 annotated data samples indicating whether they are vulgar.

OSACT5 [Mubarak et al., 2022] dataset consists of 12,698 Arabic tweets collected between June 2016 and November 2017, labeled for offensiveness and fine-grained hate speech types using emojis commonly found in offensive communications, providing a resource for offensive and hate speech detection and classification tasks. The dataset consists of three subtasks: offensiveness detection, hate speech detection, and fine-grained hate speech detection. We used 2,541 samples for each of these tasks.

ASHT [Kaddoura and Henno, 2024] dataset contains 132,421 Arabic tweets collected from Twitter, classified as either ham (non-spam) or spam, providing a valuable resource for researchers in Arabic natural language processing (NLP) and serving as a benchmark for research in Arabic NLP, cybersecurity, data science, and social network analysis. We used a subset of 1,000 samples for the spam detection section.

### B.2.2 Bengali

TRAC2020 [Bhattacharya et al., 2020] dataset is a multilingual annotated corpus of social media comments, encompassing misogynistic and aggressive comments in Indian English, Hindi, and Indian Bangla. It consists of over 20,000 comments and is annotated at two levels - aggression (overtly aggressive, covertly aggressive, and non-aggressive) and misogyny (gendered and non-gendered). Baseline experiments were conducted to develop misogyny classifiers for the three languages. TRAC2020 consists of two tasks: Aggression Detection and Misogynistic Aggression Detection. We utilized 1,000 data samples for each of Task 1 and Task 2.

BAD [Sharif and Hoque, 2022] dataset is a novel Bengali aggressive text dataset (called 'BAD') with two-level annotation, designed to identify and classify aggressive content in Bengali language. It achieves high accuracy through a weighted ensemble technique and outperforms other machine learning and deep learning baselines, with a weighted f1-score of 93.43% for identification and 93.11% for categorization tasks. We utilized a subset of one thousand data samples as the Offensive dataset.

Hate Speech [Romim et al., 2021] dataset consists of 30,000 social media user comments, covering seven categories including sports, entertainment, religion, politics, crime, celebrities, TikTok, and memes. It has been annotated through crowdsourcing and expert validation for research purposes in detecting hate speech in Bengali language. The dataset also provides benchmark experimental results for multiple deep learning models and pre-trained Bengali word vectors. We utilized 1,000 data samples from the dataset for Hate Detection.

BACD [aimansnigdha, 2018] dataset is a dataset for the Bengali language, consisting of a total of 10,200 data points with annotations for toxic, threat, obscene, insult, and racism labels. We utilized 1,000 data points from this dataset for Threat Detection and Bias Detection tasks respectively.

### B.2.3 Chinese

CCS [Jiang et al., 2019] dataset consists of two real-world spam datasets: one is an SMS dataset, and the other is a product review dataset. Both datasets were manually labeled by professionals as spam or regular emails, and their sizes and label distributions were summarized. We utilized 1000 data samples from this dataset for Spam Detection.

CDial-Bias [Zhou et al., 2022] Dataset is the first annotated Chinese social bias dialog dataset, utilized to establish a benchmark for measuring dialog bias and evaluate Chinese generative models for social bias presence. We utilized 1000 data samples from it for bias detection.

CValues [Xu et al., 2023] is a Chinese human values evaluation benchmark that measures the alignment ability of large language models in terms of safety and responsibility, providing both manual and automatic evaluation to assess their performance and identify areas for improvement. We utilized 1712 data samples from the dataset for Stance detection.

### B.2.4 English

SOLID [Rosenthal et al., 2020] dataset is an expanded dataset containing over nine million English tweets labeled in a semi-supervised fashion. It significantly improves the performance of identifying specific types and targets of offensive language when combined with the OLID dataset, particularly at lower levels of the offensive language taxonomy. We utilized 1,000 data points from the dataset for Offensive Detection.

MLMA [Ousidhoum et al., 2019] dataset is a new multilingual multi-aspect hate speech analysis dataset, which is used to evaluate state-of-the-art multilingual multitask learning approaches and improve hate speech detection and classification in general. We utilized 1000 data samples from the dataset for Hate Detection and Toxicity Detection respectively.

HOF [Davidson et al., 2017] dataset uses crowd-sourcing to collect tweets containing hate speech keywords and employs a multi-class classifier to distinguish between tweets containing hate speech, only offensive language, and those with neither. It addresses the challenge of automatically detecting hate speech on social media while separating it from other instances of offensive language. We used a subset of 1000 data samples for Hate Detection.

JMT [Kaggle, 2019] dataset is a machine learning dataset designed to identify toxic comments in online conversations, aiming to build models that can filter out rude, disrespectful, or potentially conversation-disrupting comments to create a safer and more collaborative internet environment. We used 1000 data samples each from the Threat Detection and Toxicity Detection datasets.

### B.2.5 Germany

GermEval2018 [Wiegand et al., 2018] dataset is used for identifying offensive language in German tweets, including both coarse-grained binary classification tasks and fine-grained multi-class classification tasks. We used 3,531 data points for Offensive Detection.

IWG [Ross et al., 2016] dataset aims to assess the feasibility of reliably annotating hate speech and explore the consistency between existing definitions and subjective ratings. The results indicate low reliability in users' judgments of hate speech, suggesting a need for more detailed annotation instructions. Each data instance in the dataset was annotated by two experts, and we selected 469 instances with annotations from both experts for Hate Detection, denoted as IWG_1 and IWG_2 respectively.

HASOC2020 [HASOC, 2020] dataset is a multilingual research forum and data challenge that offers tasks for identifying problematic content in English, German, and Hindi. It consists of over 10,000 annotated tweets from Twitter, and includes both coarse-grained and fine-grained classification tasks. We utilized a subset of 850 German language data from the HASOC dataset for Hate Detection.

Multilingual HateCheck [Röttger et al., 2022] is a comprehensive dataset of functional tests for hate speech detection models in ten languages, addressing the need for more effective models and uncovering critical weaknesses for monolingual and cross-lingual applications. We utilized 1000 data points from the German section of the dataset for Hate Detection.

### B.2.6 Korean

K-MHaS [Lee et al., 2022] is a multi-label dataset consisting of 109k utterances from Korean news comments, designed for hate speech detection. It effectively handles Korean language patterns, provides multi-label classification with 1 to 4 labels, and considers subjectivity and intersectionality. Strong baseline experiments using Korean-BERT-based language models show that KR-BERT with a sub-character tokenizer performs the best by recognizing decomposed characters in each hate speech class. We utilized 1000 data samples from the dataset for Hate Detection.

HateSpeech [Moon et al., 2020] dataset is a collection of 9.4K manually labeled entertainment news comments in Korean, aimed at identifying toxic speech, social bias, and hate speech. It provides benchmarks using CharCNN, BiLSTM, and BERT models, with BERT achieving the highest performance. The dataset is made publicly available and open for competitions. We utilized 1000 data samples from the dataset for Hate Detection.

HateSpeech2 [daanVeer, 2020] dataset was created by the Natural Language Processing Laboratory (NLP) at Korea National University and it includes the original dataset, a vocabulary of offensive

language, annotations, and dataset examples. The dataset is used for labeling malicious comments and has been built with word embeddings. We utilized 1000 data samples from the dataset for Hate Detection.

AbuseEval [Caselli et al., 2020] is a newly created dataset that addresses issues in annotating offensive and abusive language, specifically considering the degree of explicitness, target presence, and contextual interaction across different abusive language phenomena. We utilized 1000 data samples from the dataset for Abusive Detection.

CADD [Song et al., 2021] is a comprehensive dataset for detecting abusive language in English Reddit posts, featuring multifaceted labels and contextual information, collected through large-scale crowdsourcing and yielding meaningful performance with state-of-the-art language models. We utilized 1000 data samples from the dataset for Abusive Detection.

Waseem [Waseem and Hovy, 2016] dataset, based on critical race theory, provides annotations for over 16k tweets and aims to detect hate speech on social media by analyzing linguistic features, extra-linguistic features, and a dictionary of the most indicative words in the data. We utilized 1000 data samples from the dataset for Abusive Detection.

### B.2.7 Portuguese

OffComBR [de Pelle and Moreira, 2017] dataset is an annotated collection of offensive comments in Portuguese, gathered from news comment sections on the Brazilian web. It serves the purpose of classifying user-generated text as either positive or negative, providing a baseline for future research on the topic of hate speech detection in Portuguese. We utilized 1250 data samples from this dataset for offensive detection.

HateBR [Vargas et al., 2022] dataset is the first large-scale expert annotated corpus of Brazilian Instagram comments, specifically collected from politicians' accounts, providing binary/offensiveness-level classification and nine hate speech groups, outperforming the current state-of-the-art for Portuguese language offensive language and hate speech detection. We utilized 1000 data samples from this dataset for offensive detection.

ToLD-Br [Leite et al., 2020] is a large-scale dataset for Brazilian Portuguese, consisting of annotated tweets categorized as toxic or non-toxic, aiming to detect and prevent the proliferation of toxicity in social media, addressing the need for multilingual approaches and models aware of different categories of toxicity. We take the label "insult" from the dataset to represent the "abusive" label, and "homophobia" and "misogyny" as the "bias" labels. We have selected 1000 data samples for Abusive Detection, 1000 samples for Bias Detection, and 1000 samples for Bias Detection.

### B.2.8 Spanish

AMI [Fersini et al., 2018] dataset is a collection of Spanish and English tweets used for identifying misogyny, categorizing misogynistic behavior, and classifying targeted individuals, with contributions from multiple teams and countries. We used 1000 Spanish language data for offensive detection.

MEX-A3T [Álvarez-Carmona et al., 2018] dataset, from the track at IberEval 2018, comprises Mexican Spanish tweets and focuses on two tasks: author profiling, which aims to identify the residence and occupation of Twitter users, and aggressiveness detection, to distinguish between aggressive and non-aggressive tweets. This dataset was created specifically for these tasks and was analyzed and compared in a paper discussing the participants' results. We used 1000 data samples for offensive detection.

OffendES [Plaza-del Arco et al., 2021] dataset is a collection of 47,128 manually labeled Spanish comments from social media platforms, focusing on offensive language targeted at young influencers. It provides pre-defined offensive categories and includes confidence scores, enabling both multi-class classification and multi-output regression studies. We used 1000 data samples for offensive detection.

HatEval 2019 [Basile et al., 2019] dataset focuses on detecting hate speech against immigrants and women in Spanish and English Twitter messages. It includes two classification tasks: identifying the presence of hate speech and distinguishing between individual and group targets. HatEval was a popular SemEval-2019 task with numerous submissions and participant system analysis. We used 1000 data samples for hate detection.

HaterNet [Pereira-Kohatsu et al., 2019] dataset is an intelligent system used for monitoring and visualizing hate speech on Twitter. It provides a novel public dataset of Spanish hate speech, consisting of 6,000 expert-annotated tweets. We used 1000 data samples for hate detection.

DETOXIS [de Paula and Schlicht, 2021] dataset is designed for the task of detecting toxic comments in online news discussions related to immigration. It includes toxicity detection and toxicity level detection. Participating teams achieved good results using the BERT model on this dataset. We classified them into tags such as stereotype, improper, abusive, mockery, aggressiveness, and stance, and selected 1000 data samples for each category for Bias detection, Abusive detection, Aggressiveness detection, and Stance detection.

### B.2.9 Turkish

SemEval-2020 [Zampieri et al., 2020] provided a new, large-scale semi-supervised training dataset of over nine million English tweets and expanded the task to include four new languages, allowing for cross-lingual training and analysis. We used 3528 data samples in Turkish for Offensive Detection.

OffenseCorpus [Çöltekin, 2020] is a corpus of Turkish offensive language, comprising randomly sampled micro-blog posts from Twitter. It contains 36,232 tweets collected over an 18-month period from April 2018 to September 2019. We used 1000 data samples for Offensive Detection.

OffenseKaggle [Kaggle, 2021] Dataset is a collection of Turkish tweets from Twitter, with around 40% of them containing offensive or vulgar content. We used 1000 data samples for Offensive Detection.

OffenseKaggle_2 [Kaggle, 2022] dataset is an enhanced version of an existing offensive language research dataset, which has been expanded and annotated using contextual data mining techniques. It addresses the issue of class imbalance in existing studies and provides a more comprehensive and robust dataset for Turkish offensive language detection tasks. We used 1000 data samples for Offensive Detection.

ATC [Karayiğit et al., 2021] dataset is a publicly available dataset for detecting abusive Turkish comments on Instagram. It consists of 10,528 abusive and 19,826 non-abusive comments, with sentiment annotations at the sentence level. We used 1000 data samples for Offensive Detection.

Turkish Spam [mis, 2019] dataset contains both spam and normal emails written in Turkish. A total of 330 spam emails and 496 normal emails were collected from several personal accounts. We used 825 pieces of data for spam detection.

OffenseCorpus [Çöltekin, 2020] dataset is a large collection of Turkish offensive language from Twitter micro-blog posts, annotated based on recent practices. It includes 36,232 randomly sampled tweets from April 2018 to September 2019, with 19% containing offensive language. We used 1000 of the data for Finegrained offensive detection.

## C   Evaluation Metrics and Prompts

### C.1   Evaluation metrics

#### C.1.1   Perplexity

The perplexity on a test dataset $D$ and a language model $\mathcal{M}$ is computed as:

$$\mathrm{ppl}(D, \mathcal{M}) = \exp\left(-\frac{1}{N}\sum_{i=1}^{N} \log P(x_i|\mathcal{M})\right),$$

where $N$ represents the total number of tokens in $D$, $x_i$ represents the $i$-th token in the test dataset, $P(x_i|\mathcal{M})$ represents the probability of generating token $x_i$ given the model $\mathcal{M}$, and $\log$ is the natural logarithm.

In usual, a lower perplexity value indicates better performance of the model on the test data. However, for evaluating the data quality to train model, a higher perplexity value means it can bring more valuable information.

Table 5: Prompts for evaluation.

| Task | Prompt |
|------|--------|
| offensive_detect | if the following sentence is offensive, just say \"OFF\", otherwise, say \"NOT_OFF\" without explanation: |
| hate_detect | if the following sentence has hate speech, just say \"HS\", otherwise, say \"NOT_HS\" without explanation: |
| vulgar_detect | if the following sentence has vulgar speech, just say \"Vulgar\", otherwise, say \"NOT_Vulgar\" without explanation: |
| spam_detect | if the following sentence is spam tweet, just say \"Spam\", otherwise, say \"NOT_Spam\" without explanation: |
| stereotype_detect | if the following sentence has stereotype speech, just say \"1\", otherwise, say \"0\" without explanation: |
| mockery_detect | if the following sentence has mockery speech, just say \"1\", otherwise, say \"0\" without explanation: |
| insult_detect | if the following sentence has insult speech, just say \"1\", otherwise, say \"0\" without explanation: |
| improper_detect | if the following sentence has improper speech, just say \"1\", otherwise, say \"0\" without explanation: |
| aggressiveness_detect | if the following sentence has aggressiveness speech, just say \"1\", otherwise, say \"0\" without explanation: |
| toxicity_detect | if the following sentence has toxicity speech, just say \"1\", otherwise, say \"0\" without explanation: |
| negative_stance_detect | if the following sentence has negative stance speech, just say \"1\", otherwise, say \"0\" without explanation: |
| homophobia_detect | if the following sentence has homophobia speech, just say \"1\", otherwise, say \"0\" without explanation: |
| racism_detect | if the following sentence has racism speech, just say \"1\", otherwise, say \"0\" without explanation: |
| misogyny_detect | if the following sentence has misogyny speech, just say \"1\", otherwise, say \"0\" without explanation: |
| threat_detect | if the following sentence has threat speech, just say \"1\", otherwise, say \"0\" without explanation: |
| bias_on_gender_detect | if the following speech expressing bias on gender, just say \"1\", otherwise, say \"0\" without explanation: |
| hostility_directness_detect | if the following speech expressing hostility directness, just say \"1\", otherwise, say \"0\" without explanation: |
| hate_offens_detect | if the following sentence contains hate speech, just say \"0\", else if contains offensive language, say \"1\", otherwise, say \"2\" without explanation: |
| hate_detect_fine-grained | if the following sentence doesn't have hate speech, just say \"NOT_HS\", otherwise, label the hate speech with \"HS1\"(Race), \"HS2\"(Religion), \"HS3\"(Ideology), \"HS4\"(Disability), \"HS5\"(Social Class), \"HS6\"(Gender) without explanation: |
| offensive_detect_finegrained | if the following sentence doesn't have offensive speech, just say \"non\", otherwise, label the offensive speech with \"prof\"(profanity, or non-targeted offense), \"grp\"(offense towards a group), \"indv\"(offense towards an individual), \"oth\"(ffense towards an other (non-human) entity, often an event or organization) without explanation: |

## C.1.2 Diversity Gain

We use the diversity gain [Bilmes, 2022] to measure what extent can our generated dataset bring data diversity to the base dataset. The base dataset can be defined as $\mathcal{D}_{base} = \{x_i = (q_i, r_i, a_i)\}_{i=1}^{N}$ with $N$ samples. The new generated dataset is defined as $\mathcal{D}_{new} = \{x_i = (q_i, r_i, a_i)\}_{i=1}^{M}$ with $M$ samples. And the diverse gain of $D_{new}$ relative to $D_{base}$ can be expressed as:

$$d_{gain} = \frac{1}{M} \sum_{x_i \in \mathcal{D}_{new}} \min_{x_j \in \mathcal{D}_{base}} (\|\mathbf{f}(\mathbf{x_i}) - \mathbf{f}(\mathbf{x_j})\|),$$

where $f$ is the feature extractor, and we use OpenAI Embedding API text-embedding-ada-002 to extract features.

## C.2 Prompts for evaluation

For evaluation, we input a system prompt to induce LLM to specific culture. For example, we add "You are an Arabic chatbot that knows Arabic very well." as system prompt for evaluation on Arabic understanding. As for the prompt for evaluation on different tasks, the detailed prompts are shown in Table 5.

**Prompts for open-ended tasks:**

1. Prompt for generate open-ended dataset:
   Can you generate some open-ended question according to the following questions:
   {Some questions on the same topic}

2. Prompt for evaluate open-ended dataset:
   Question: {q}
   Answer A: {ans 1}
   Answer B: {ans 2}
   Comparing these two answers, which answer is more socially aligned with country culture?
   Options:
   A. Answer A.
   B. Answer B.

3. Prompt for rating sentences similarity
   Rate the semantic similarity of two input sentences on a scale of 1 - definitely not to 5 - perfectly.
   Sentence 1: {item 1}
   Sentence 2: {item 2}

Table 6: Information retrieved via RAG

| Culture | Information |
| --- | --- |
| Arabic | Arab culture is the culture of the Arabs, from the Atlantic Ocean in the west to the Arabian Sea in the east, in a region of the Middle East and North Africa known as the Arab world. The various religions the Arabs have adopted throughout their history and the various empires and kingdoms that have ruled and took lead of the civilization have contributed to the ethnogenesis and formation of modern Arab culture. Language, literature, gastronomy, art, architecture, music, spirituality, philosophy and mysticism are all part of the cultural heritage of the Arabs. |
| Bengali | The culture of Bengal defines the cultural heritage of the Bengali people native to eastern regions of the Indian subcontinent, mainly what is today Bangladesh and the Indian states of West Bengal and Tripura, where they form the dominant ethnolinguistic group and the Bengali language is the official and primary language. Bengal has a recorded history of 1,400 years. The Bengalis are dominant ethnolinguistic group. The Bengal region has been a historical melting point, blending indigenous traditions with cosmopolitan influences from pan-Indian subcontinental empires. Dhaka (Dacca) became the capital of Mughal Bengal (Bengal Subah) and the commercial (financial) capital (1610-1757) of Mughal India. Dhaka is the largest and richest Bengali (Bangali) mega city in the world and also the 3rd largest and richest mega city in (Indian sub continent) after Mumbai (Bombay or MMR) and Delhi (NCR). Dhaka is a Beta Global City (Moderate Economic Centre). As a part of the Bengal Presidency, Bengal also hosted the region's most advanced political and cultural centers during British rule. |
| Chinese | Chinese culture is one of the world's oldest cultures, originating thousands of years ago. The culture prevails across a large geographical region in East Asia with Sinosphere in whole and is extremely diverse, with customs and traditions varying greatly between counties, provinces, cities, towns. The terms 'China' and the geographical landmass of 'China' have shifted across the centuries, before the name 'China' became commonplace in modernity. Chinese civilization is historically considered a dominant culture of East Asia. With China being one of the earliest ancient civilizations, Chinese culture exerts profound influence on the philosophy, virtue, etiquette, and traditions of Asia. Chinese characters, ceramics, architecture, music, dance, literature, martial arts, cuisine, arts, philosophy, etiquette, religion, politics, and history have had global influence, while its traditions and festivals are celebrated, instilled, and practiced by people around the world. |
| English | The culture of the United States of America, also referred to as American culture, encompasses various social behaviors, institutions, and norms in the United States, including forms of speech, literature, music, visual arts, performing arts, food, sports, religion, law, technology as well as other customs, beliefs, and forms of knowledge. American culture has been shaped by the history of the United States, its geography, and various internal and external forces and migrations. Several historical ethnicities make up American culture: Yankee (Anglo-American), Tejano, Louisiana French (Cajun, Creole), Pennsylvania Dutch (Fancy Dutch, Amish), New York Dutch, Texas German, Alaskan Russian, Puerto Rican, Hawaiian. |
| Germany | The culture of Germany has been shaped by major intellectual and popular currents in Europe, both religious and secular. German culture originated with the Germanic tribes, the earliest evidence of Germanic culture dates to the Jastorf culture in Northern Germany and Denmark. Contact with Germanic tribes were described by various Greco-Roman authors. The first extensive writing done on Germanic culture can be seen during the Roman Imperial Period with Germania by Tacitus. |
| Korean | The contemporary culture of South Korea developed from the traditional culture of Korea which was prevalent in the early Korean nomadic tribes. By maintaining thousands of years of ancient Korean culture, with influence from ancient Chinese culture, South Korea split on its own path of cultural development away from North Korean culture since the division of Korea in 1948. The industrialization, urbanization and westernization of South Korea, especially Seoul, have brought many changes to the way Korean people live. Changing economics and lifestyles have led to urbanization—a concentration of population in major cities (and depopulation of the rural countryside), with multi-generational households separating into nuclear family living arrangements. Today, many cultural elements from South Korea, especially popular culture, have spread across the globe and have become some of the most prominent cultural forces in the world. |
| Portuguese | The culture of Portugal is a very rich result of a complex flow of many different civilizations during the past millennia. From prehistoric cultures, to its Pre-Roman civilizations (such as the Lusitanians, the Gallaeci, the Celtici, and the Cynetes, amongst others), passing through its contacts with the Phoenician-Carthaginian world, the Roman period (see Hispania, Lusitania and Gallaecia), the Germanic invasions of the Suebi, Buri (see Kingdom of the Suebi) and Visigoths (see Visigothic Kingdom), Viking incursions, Sephardic Jewish settlement, and finally, the Moorish Umayyad invasion of Hispania and the subsequent expulsion, during the Reconquista, all have made an imprint on the country's culture and history. The name of Portugal itself reveals much of the country's early history, stemming from the Roman name Portus Cale, a Latin name meaning \"Port of Cale\" (Cale likely is a word of Celtic origin - Cailleach-Bheur her other name; the Mother goddess of the Celtic people as in Calais, Caledonia, Beira. She was the one who, with a hammer created mountains and valleys; the one who hid in stones and trees - Mother nature), later transformed into Portucale, and finally into Portugal, which emerged as a county of the Kingdom of León see County of Portugal) and became an independent kingdom in 1139. During the 15th and 16th centuries, Portugal was a major economic, political, and cultural power, its global empire stretching from the Americas, to Africa, and various regions of Asia and Oceania. Portugal, as a country with a long history, is home to several ancient architectural structures, as well as typical art, furniture and literary collections mirroring and chronicling the events that shaped the country and its peoples. It has a large number of cultural landmarks ranging from museums to ancient church buildings to medieval castles, which testify its rich national cultural heritage. Portugal is home to fifteen UNESCO World Heritage Sites, ranking it 8th in Europe and 17th in the world. |
| Spanish | The culture of Spain is influenced by its Western origin, its interaction with other cultures in Europe, its historically Catholic religious tradition, and the varied national and regional identities within the country. It encompasses literature, music, visual arts, cuisine as well as contemporary customs, beliefs, institutions, and social norms. Beyond Spain, Spanish culture is the foundation of most of Latin American cultures and the Filipino culture. The ancient peoples of Spain included Tartessians, Celts, Iberians, Celtiberians, Phoenicians as well as Greek colonies. Spain largely came under the rule of Carthage and was then entirely conquered by Rome, becoming a province of the Roman empire. The name of Spain derives from the Latin term Hispania, itself a name of Punic origin. In the areas of language and religion, the ancient Romans left a lasting cultural, legal and administrative legacy in the Spanish history. The subsequent course of Spanish history added new elements to the country's culture and traditions. The Visgoths established a united Hispania and kept the Latin and Christian legacy in Spain between the fall of the Roman Empire and the Early Middle Ages. Muslim influences played a significant role during the Middle Ages in the areas conquered by the Umayyads. However, these influences were not completely assimilated into the Spanish culture, leading to conflicts and ultimately to the Christian Reconquista that would largely shape the culture of the country. |
| Turkish | The culture of Turkey (Turkish: Türkiye kültürü) or the Turkish culture (Türk kültürü) combines a heavily diverse and heterogeneous set of elements that have been derived from the various cultures of the Eastern European, Eastern Mediterranean, Caucasian, Middle Eastern and Central Asian traditions. Many of these traditions were initially brought together by the Ottoman Empire, a multi-ethnic and multi-religious state spanning across Southern Europe, Eastern Europe, the Middle East and North Africa. During the early years of the Republic of Turkey, established after the collapse of the Ottoman Empire, the government invested large sums of resources into fine arts such as architecture and sculpture, and other artistic fields around the country in-line with the newly implemented reformist and West-leaning policies. This was done as part of a process of modernization, westernization, and of creating and outlining a new Turkish cultural identity, rather than the previously established and depicted Ottoman identity. |

## C.3   Prompts for RAG

The detailed information retrieved via RAG is shown in Table 6.

# D   Experimental Results

## D.1   Detailed results

The detailed results are shown in Figure 6.

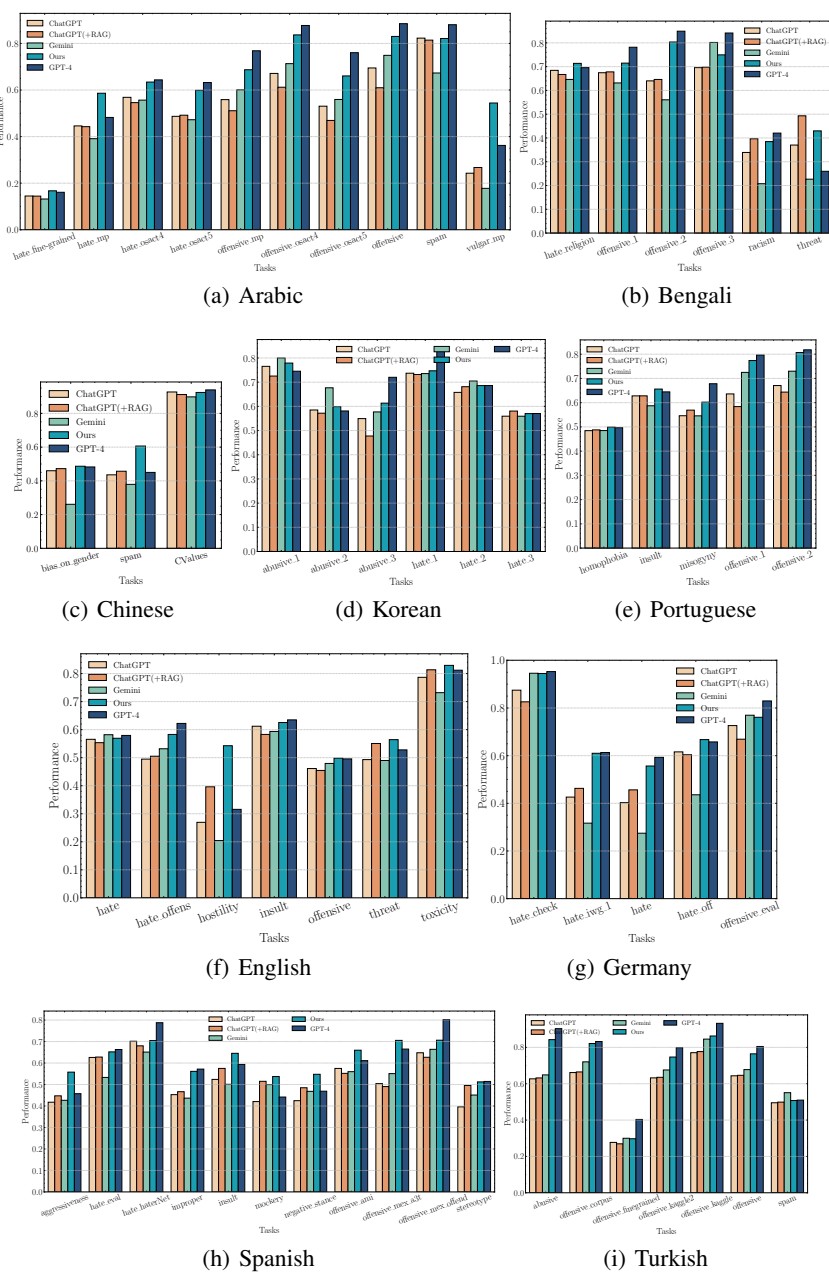

Figure 6: Detailed results on all tasks and all cultures.

Table 7: Comparison with the latest cultural specific LLMs.

| Chinese | Bias | Spam | Avg |
|---|---|---|---|
| SeaLLM | .237 | .357 | .297 |
| Taiwan_LLM | .446 | .341 | .394 |
| CultureLLM _One | .469 | .602 | .536 |
| CultureLLM _Zh | **.487** | **.606** | **.547** |

| Arabic | Hate | Offensive | Avg |
|---|---|---|---|
| CultureBank | .540 | .642 | .591 |
| CultureLLM _One | .540 | .571 | .555 |
| CultureLLM _Ko | **.576** | **.661** | **.618** |

| Korean | Abusive | Hate | Avg |
|---|---|---|---|
| SeaLLM | .523 | .474 | .499 |
| CultureBank | .635 | .522 | .579 |
| CultureLLM _One | .619 | .504 | .561 |
| CultureLLM _Ko | **.664** | **.628** | **.646** |

## D.2 Comparative analysis with other cultural specific models

We have conducted a comparative analysis with models designed for cultural understanding. Given the absence of models specifically tailored to Arabic, Bengali, German, Portuguese, Spanish, and Turkish cultures, our analysis focused on models with proficiency in Chinese and Korean cultures. Specifically, we examined SeaLLM [Nguyen et al., 2023b], which targets Southeast Asian cultural nuances, TAIWAN LLM [Lin and Chen, 2023], which is dedicated to Chinese cultural contexts, and CultureBank [Shi et al., 2024], which focuses on several cultural groups. Table 7 highlight our findings in the Chinese and Korean benchmarks. The results clearly demonstrate that our CultureLLMs significantly outperform both SeaLLM and TAIWAN LLM, showcasing their superior ability to understand cultural subtleties on a broader scale.

## D.3 Results analysis on WVS seed data

We analyze the relevance of each task with the WVS and investigate if it correlates with the experimental results.

**Offensive language detect:**

1. Cultural Context and Sensitivity to Offensive Language: The World Values Survey aims to capture cultural values and beliefs across different societies. One aspect of cultural values is the tolerance or acceptance of offensive language. In some cultures, certain words or expressions may be considered highly offensive, while in others they may be more tolerated or even commonly used.

2. Social Norms and Acceptance: The survey may reveal societal attitudes towards the use of offensive language in various contexts, such as in public discourse, media, or interpersonal communication.

**Hate speech detect:**

1. Societal Norms and Attitudes: The WVS provides data on societal norms, attitudes towards minorities, and levels of societal trust. This data can help understand the underlying societal conditions that might foster hate speech or, conversely, promote tolerance and inclusivity.

2. Cultural Context: Understanding the cultural context is crucial for effectively detecting and interpreting hate speech. The WVS offers a rich dataset for understanding cultural differences in values and norms, which can inform more nuanced hate speech detection algorithms that are sensitive to context and do not inadvertently suppress legitimate expressions of cultural or political dissent.

**Stance detect:**

1. Understanding Contextual Influences on Stance: The WVS can provide the cultural and societal background needed to understand why certain stances are more prevalent in specific regions or among certain demographic groups. This context can be invaluable for interpreting the results of stance detection analyses, especially when comparing stances across different cultures and societies.

**Toxicity detect:**

1. Reflection of Societal Norms in Online Behavior: The WVS provides insights into the prevailing norms and values within societies, which can indirectly inform the context within which toxic behavior manifests online. Understanding societal attitudes towards diversity, authority, individual freedom, and tolerance can help in interpreting the root causes of toxic behavior and devising appropriate responses.

2. Injection of More cultural nuances: Data from the WVS can inject more information that sensitive to cultural nuances and differences in value systems. This can prevent the misclassification of content that may be culturally specific or context-dependent, reducing the risk of censoring legitimate expressions of cultural or political identity.

**Threat detect:**

1. Understanding Motivations and Behaviors: Insights from the WVS can help understand the cultural and societal contexts that may influence the behavior of individuals or groups posing threats. This knowledge can inform more targeted and effective threat detection and mitigation strategies that consider the root causes of conflict or aggression.

2. Cultural Sensitivity in Security Measures: Incorporating findings from the WVS can lead to more culturally sensitive security practices that respect local values and norms. This is crucial in global operations where misunderstanding cultural nuances can lead to ineffective or counterproductive security measures.

**Bias detect:**

1. Understanding Societal Norms and Attitudes: Insights from the WVS can help in understanding the cultural and societal norms that underlie biases. By analyzing patterns in global values and beliefs, we can identify prevalent stereotypes, prejudices, and discriminatory attitudes that may need to be addressed in bias detection efforts.

2. Injection of More cultural nuances: The WVS data can provide valuable context that are sensitive to cultural differences in values and norms. This is better equipped to detect and mitigate biases in data sets that reflect cultural nuances, ensuring that AI-driven decisions are fair and equitable across different societal contexts.

**Abusive detect:**

1. Cultural Contexts of Abuse: The WVS can help identify cultural norms that influence perceptions of what constitutes abusive behavior. This is crucial for developing detection systems that are sensitive to cultural differences, ensuring that they can effectively identify abuse without mistakenly flagging culturally specific but non-abusive interactions.

2. Injection of More cultural nuances: Insights from the WVS can inform the development of more nuanced algorithms for detecting abusive behavior by providing context on societal values and norms. This can help in training models to recognize the subtle nuances that differentiate abusive from non-abusive communication in different cultural settings.

3. Evaluating Tolerance Levels: The WVS data can provide insights into societal tolerance levels towards different forms of behavior, including what might be considered abusive. This can help in assessing the urgency and type of interventions needed to address abusive behaviors in various cultural contexts.

**Spam detect:**

1. Cultural Variations in Communication: The WVS can shed light on cultural differences in communication styles and preferences, which can inform more nuanced spam detection

Table 8: Ablation study for CultureLLM-One.

| | Ar | Bn | Zh | En | De | Ko | Pt | Es | Tr | avg |
|---|---|---|---|---|---|---|---|---|---|---|
| GPT-3.5 | 0.517 | 0.567 | 0.448 | 0.526 | 0.609 | 0.642 | 0.593 | 0.517 | 0.587 | 0.556 |
| +WVS | 0.543 | 0.583 | 0.501 | 0.568 | 0.641 | 0.531 | 0.603 | 0.560 | 0.590 | 0.569 |
| +WVS+a | 0.532 | 0.572 | 0.482 | 0.543 | 0.631 | 0.563 | 0.620 | 0.543 | 0.585 | 0.564 |
| +WVS+b | 0.543 | 0.571 | 0.505 | 0.543 | 0.621 | 0.536 | 0.598 | 0.559 | 0.590 | 0.563 |
| CultureLLM-One | 0.583 | 0.597 | 0.536 | 0.589 | 0.662 | 0.571 | 0.627 | 0.596 | 0.609 | 0.597 |

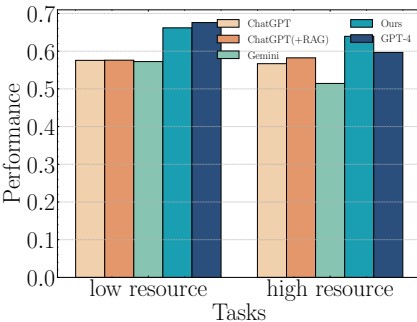

Figure 7: We compare CultureLLM with baselines on low-resource language tasks and high-resource language tasks.

algorithms that are better able to distinguish between legitimate mass communications and spam in different cultural contexts.

2. Attitudes Towards Technology and Privacy: Insights from the WVS regarding societal attitudes towards technology use, privacy, and data protection can help in tailoring spam detection efforts to respect cultural norms and expectations. For instance, societies with a high value on privacy might be more receptive to stringent spam filters.

3. Attitudes Towards Technology and Privacy: Insights from the WVS regarding societal attitudes towards technology use, privacy, and data protection can help in tailoring spam detection efforts to respect cultural norms and expectations. For instance, societies with a high value on privacy might be more receptive to stringent spam filters.

## D.4 Ablation on CultureLLM-One

The ablation study on CultureLLM-One is shown in Table 8.

## D.5 Analysis on low-resource language tasks and high-resource language tasks

Figure 7 shows the comparison between low and high resource tasks.

## D.6 Results on other low-resource cultures

We fine-tuned a CultureLLM-el model specifically tailored for the Greek culture and assessed its efficacy on two Greek benchmarks focused on offensive language detection. Presented in the table below are the outcomes obtained from Zampieri et al. [2020], Pitenis et al. [2020]. Our findings indicate that, on average, CultureLLM-el surpasses gpt-3.5-turbo [OpenAI, 2023a] by 5.6% in performance. The results are shown in Table 9.

Table 9: Results on Greek culture

| Model | OffensEval2020 | gazzetta |
|---|---|---|
| GPT-3.5-turbo | 0.4773 | 0.3771 |
| CultureLLM-el | 0.5201 | 0.4461 |

## D.7  Case study

Figure 8 show the responses from CultureLLM and ChatGPT on four different cultures.

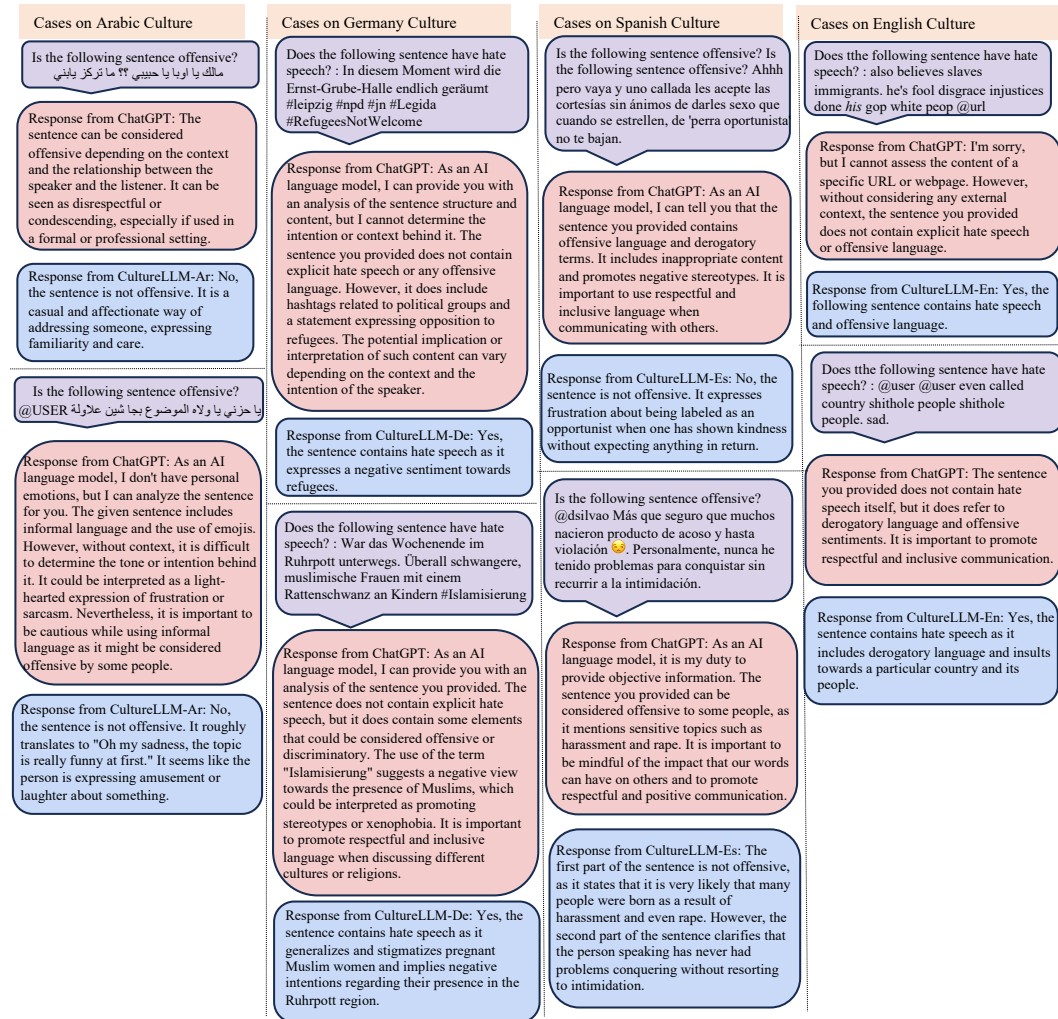

Figure 8: The responses from ChatGPT and CultureLLM on four different cultures

# E  Fine-tuning on Llama and Results

## E.1  Setup

We use Lora [Hu et al., 2021] to fine-tune Llama-70b-Chat. The setting for Lora are list below:

- lora_alpha: 16
- lora_dropout: 0.1
- r: 64
- bias: none
- task_type: CAUSAL_LM

The detailed setting for training are list below:

- num_train_epochs: 6

Table 10: More results on fine-tuning using Llama-2-70b model.

|  | Ar | Bn | Zh | En | De | Ko | Pt | Es | Tr | AVG |
|---|---|---|---|---|---|---|---|---|---|---|
| Llama | 0.4852 | 0.3202 | 0.3315 | 0.4108 | 0.3631 | 0.4869 | 0.3792 | 0.3554 | 0.3856 | 0.3909 |
| CultureLLM-one | 0.4911 | 0.3212 | 0.3533 | 0.4211 | 0.3827 | 0.4923 | 0.3892 | 0.3611 | 0.3855 | 0.3997 |
| CultureLLM | 0.5047 | 0.3398 | 0.3631 | 0.4214 | 0.3925 | 0.5019 | 0.3956 | 0.3752 | 0.3899 | 0.4093 |

Table 11: Results on forgetting experiments on Llama-2-70b

| Model | GSM8K | BBH |
|---|---|---|
| Llama-2-70b | 56.8 | 51.2 |
| CultureLLM-Ar | 56.9 | 53.2 |
| CultureLLM-Bn | 56.8 | 52.1 |
| CultureLLM-Zh | 56.8 | 56.6 |
| CultureLLM-En | 56.8 | 48.8 |
| CultureLLM-De | 56.9 | 51.8 |
| CultureLLM-Ko | 56.8 | 53.6 |
| CultureLLM-Pt | 56.8 | 55.1 |
| CultureLLM-Es | 56.7 | 53.2 |
| CultureLLM-Tr | 56.8 | 55.5 |
| Avg | 56.8 | 53.3 |

- er_device_train_batch_size: 4
- gradient_accumulation_steps: 1
- optim: paged_adamw_32bit
- learning_rate: 2e-4
- weight_decay: 0.001
- fp16: False
- bf16: False
- max_grad_norm: 0.3
- max_steps: -1
- warmup_ratio: 0.03
- group_by_length: True
- lr_scheduler_type: constant
- report_to: tensorboard

### E.2 Detailed results

The results of fine-tuning Llama2 are shown in Table 10. It shows the similar trends as ChatGPT's results. We conduct forgetting experiments on Llama-2. As the results shown in Table 11, CultureLLM does not bring negative effect on general tasks.

## F Details on Human Study

Information on participant in human study are shown in Table 12.

Participants are asked to rate the 100 samples according to the following criterion:

1. **Score 1:** i. The sentences convey distinctly different ideas or concepts. ii. No apparent connection or shared meaning.

2. **Score 2:** i. Limited commonality in meaning, with noticeable disparities in wording. ii. Shared concepts but with significant differences in expression.

Table 12: Information on participants in human study

| Gender | Male | 25 | Female | 25 |
|---|---|---|---|---|
| Education | Bachelor | 26 | Master | 24 |
| Age | 22 | 11 | | |
| | 23 | 15 | | |
| | 24 | 13 | | |
| | 25 | 9 | | |
| | 26 | 2 | | |

3. **Score 3:** i. Some overlap in meaning, but notable differences in wording or phrasing. ii. Context or emphasis might differ slightly.

4. **Score 4:** i. Minor variations in wording or structure, but the core meaning remains consistent. ii. Synonymous expressions and interchangeable terms are present.

5. **Score 5:** i. The sentences convey the same information using different words. ii. No discernible difference in meaning or context.

