# OpenReview forum: "CultureLLM: Incorporating Cultural Differences into Large Language Models"
_NeurIPS.cc/2024/Conference — NeurIPS 2024 poster_

### Official Review · Reviewer_eAV9 · 2024-07-04

**Soundness:** 3
**Presentation:** 3
**Contribution:** 3
**Rating:** 6
**Confidence:** 4

**Summary:**

This paper introduces a pipeline that can enhance LLM's ability to culture-aware tasks (such as hate speech detection, and bias detection. Their proposed CultureLLM included three stages: sampling, semantic data augmentation, and fine-tuning. They investigate the effectiveness of CultureLLM in nine languages and eight culture-related task. Their extensive experiments and analysis demonstrate that CultureLLM significantly improves LLM performance on culture-aware tasks while preventing catastrophic forgetting.

**Strengths:**

1. The paper presents solid experiments to investigate the effectiveness of the proposed method.
2. The paper is well-organized and easy to read.

**Weaknesses:**

1. Rely on human-annotated dataset (i.e., WVS). The author used WVS dataset as seed data to augment their fine-tuning dataset. This limits the applicability of the proposed method.
2. Most evaluation downstream tasks are anti-social detection tasks (such as offensive language, hate speech, toxicity and abusive language detections). The results may be biased to such tasks. I also wonder why these tasks are culture-related tasks. Are there any particular examples?
3. Some experiment details need clarification. 1. What are your fine-tuning hyperparamters? Like learning rate and training steps? 2. When you evaluate CultureLLM the downstream tasks, are the input samples in English or the particular language?

**Questions:**

See weakness.

More questions:
1. Did you check if any fine-tining samples are overlapped with your downstream samples? Or some samples are very similar to downstream tasks?
2. In your ablation studies, WVS+a only used a semantic template. Do you mean that LLMs are fine-tuned on semantic templates?

**Limitations:**

This proposed method uses the WVS dataset as seed data to augment their fine-tuning dataset. This limits its applicability.

---

> ### Author Rebuttal · Authors · 2024-08-05
>
> **W1: Rely on human-annotated dataset (i.e., WVS). The author used WVS dataset as seed data to augment their fine-tuning dataset. This limits the applicability of the proposed method.**
>
> Using human-annotated data to help research is a popular choice for most of the LLM papers, as shown in [1-7], where they all use human-annotated dataset as seed data. On this point, one noteworthy advantage of our approach is that we only use 50 samples, which is significantly less that existing literature, claiming our less reliance on annotations.
>
> [1] Wang, Ruida, Wangchunshu Zhou, and Mrinmaya Sachan. "Let's Synthesize Step by Step: Iterative Dataset Synthesis with Large Language Models by Extrapolating Errors from Small Models." arXiv preprint arXiv:2310.13671 (2023).
>
> [2] Chen, Zixiang, et al. "Self-play fine-tuning converts weak language models to strong language models." arXiv preprint arXiv:2401.01335 (2024).
>
> [3] Yu, Longhui, et al. "Metamath: Bootstrap your own mathematical questions for large language models." arXiv preprint arXiv:2309.12284 (2023).
>
> [4] Li, Xian, et al. "Self-alignment with instruction backtranslation." arXiv preprint arXiv:2308.06259 (2023).
>
> [5] Liu, Bingbin, et al. "Tinygsm: achieving> 80% on gsm8k with small language models." arXiv preprint arXiv:2312.09241 (2023).
>
> [6] Singh, Avi, et al. "Beyond human data: Scaling self-training for problem-solving with language models." arXiv preprint arXiv:2312.06585 (2023).
>
> [7] Huang, Yiming, et al. "Key-point-driven data synthesis with its enhancement on mathematical reasoning." arXiv preprint arXiv:2403.02333 (2024).
>
> **W2: Most evaluation downstream tasks are anti-social detection tasks (such as offensive language, hate speech, toxicity and abusive language detections). The results may be biased to such tasks. I also wonder why these tasks are culture-related tasks. Are there any particular examples?**
>
> - First of all, the downstream tasks are just evaluation, but not fine-tuning datasets. That being said, we only perform evaluation on those downstream datasets. Therefore, your comment "the results may be biased to such tasks" does not hold.
> - Second, it is actually challenging to find enough public datasets to evaluate the cultural models. Those downstream tasks are what we have found. Most of them may relate to anti-social, but our models are not. In fact, those extreme cases are what cultural misunderstanding could possibly occur, which can help "stress test" our models and verify its real application.
> - Third, the table below shows the input-output pairs of "offensive language detect" task. The offensive contents are different for different cultures. The model needs to judge if the input content is offensive and label it with "OFF" or "NOT". And the label is related to the cultural context.
>
> | Culture | Content | Label |
> | -------- | -------- | -------- |
> | Arabic     | @USER يا حُب يا دُنيا جديدة يا احلى يا احلى ابيات القصيدة♥️.     | NOT     |
> | German     | Liebe Freunde, Gauck,das Symbol des Sklaventums!    | OFF     |
> | Portuguese     | AVANTE  MEDIDAS o Brasil clama por elas pra cima dos corruptos Cadeia neles   | NOT     |
>
> **W3: Some experiment details need clarification. 1. What are your fine-tuning hyperparamters? Like learning rate and training steps? 2. When you evaluate CultureLLM the downstream tasks, are the input samples in English or the particular language?**
>
> Thanks for the reminder! We will add the details into the next version of paper.
> - Fine-tuning hyperparamters. For gpt-3.5-turbo, the number of training epochs is 3. And we use default settings for other hyperparamters. For Llama-70b-Chat, the details on setup are shown in Sec.D.1.
> - Language of input sample: The evaluation benchmarks are all in particular languages.
>
> **Q1: Did you check if any fine-tuning samples are overlapped with your downstream samples? Or some samples are very similar to downstream tasks?**
>
> We checked manually and it turns out the fine-tuning samples are not overlapped with the downstream samples. It is easy to understanding since the seed data from World Value Survey(WVS) are related to cultural value, which focus on more abstract information. The evaluation downstream tasks care about the meaning of a specific sentence.
>
> **Q2: In your ablation studies, WVS+a only used a semantic template. Do you mean that LLMs are fine-tuned on semantic templates?**
>
> No. "WVS+a" means the output of Step 2 instead of output of Step 3 in Fig 2.
>
> - - -
>
> If you are satisfied with the answers, please consider improving the rating to support our work! If you have more questions, please do not hesitate to let us know:)

---

> > ### Comment · Reviewer_eAV9 · 2024-08-10
> >
> > Thanks for your response.
> >
> > I would like to clarify my comment "The results may be biased to such tasks."
> > I understand that the evaluation datasets are not used to train the model and do not introduce any bias into the model. My concern is whether the evaluation results can comprehensively reflect the model's ability to solve culture-related tasks or just reflect its ability to detect anti-social language. For example, I think that irony and sarcasm are highly culture-related and require the model to understand the implicit expression in the utterances. Can you think about any other tasks that are culture-related?

---

> > > ### Author Response · Authors · 2024-08-10
> > > **Further Response**
> > >
> > > Thanks for your response! We answer your question about culture-related tasks from two aspects: 1) why the tasks in the paper are culture-related; and 2) other tasks.
> > >
> > > 1) Why the tasks in our paper are culture-related:
> > >
> > > - offensive language detect:
> > > Offensive language detection is culture-related because what is considered offensive varies across cultures, influenced by different norms, values, and historical contexts. Effective detection requires understanding the specific cultural context to accurately interpret language and avoid misinterpretations. Irony and sarcasm are also seen as offensive language.
> > > - hate speech detect:
> > > Hate speech detection is culture-related because cultural and historical contexts shape what is considered hateful or discriminatory. For instance, expressions of prejudice that might be seen as hate speech in one country could be viewed as acceptable or less offensive in another, such as comments about national identity or ethnic groups. Additionally, cultural attitudes toward different social groups and historical events influence the interpretation of what constitutes hate speech, making cultural awareness essential for accurate detection.
> > > - stance detect:
> > > Stance detection is culture-related because cultural context shapes how opinions, attitudes, and expressions are interpreted, influencing what is seen as supportive, neutral, or oppositional. To accurately assess stance, it's essential to understand the cultural background and nuances of the language used.
> > > - toxicity detect:
> > > Toxicity detection is culture-related because perceptions of what constitutes harmful or abusive language vary across cultures. For example, a phrase deemed disrespectful in one culture might be seen as a mild critique in another. Additionally, cultural norms around politeness and confrontation can influence how toxicity is expressed and perceived, making cultural context essential for accurate detection.
> > > - threat detect:
> > > Threat detection is culture-related because different cultures have varying thresholds for what is considered threatening or aggressive. For instance, direct confrontation or strong language might be seen as a serious threat in one culture, while in another, it could be interpreted as a standard form of assertiveness or debate. Additionally, cultural attitudes towards authority and conflict can influence how threats are expressed and understood, making it crucial to consider cultural context for accurate detection.
> > > - bias detect:
> > > For example, gender biases might be recognized differently in cultures with varying levels of gender equality, and what is considered a racial stereotype can differ across societies. Understanding these cultural nuances is essential for accurately identifying and addressing bias in language and behavior.
> > > - abusive detect:
> > > For example, humor or criticism that might be perceived as harmless in one culture could be seen as abusive in another, such as the use of sarcasm or direct criticism. Additionally, cultural differences in communication styles and social hierarchies affect how abuse is expressed and recognized, making cultural context crucial for accurate detection.
> > > - spam detect
> > > Spam detection is culture-related because different cultures have varying norms around communication and marketing practices. For example, aggressive promotional tactics that might be considered spammy in one region could be standard business practices in another, such as frequent unsolicited messages. Additionally, cultural attitudes toward privacy and advertising influence how spam is defined and identified, requiring a nuanced understanding of cultural context for effective detection.

---

> > > > ### Author Response · Authors · 2024-08-10
> > > >
> > > > 2) Other tasks that are also culture-related:
> > > >
> > > >
> > > > There are lots of culture-related tasks, such as sentiment analysis, translation and customer service automation. For sentiment analysis, emotional expression and interpretation vary across cultures. For example, in some cultures, people might express discontent indirectly or through subtle cues, making it challenging for algorithms to detect negative sentiment accurately. Additionally, cultural norms around politeness and formality can influence how sentiments are conveyed, requiring context-aware approaches to understand true sentiments. As for translation, it involves not only converting words but also adapting cultural nuances and context. For example, idiomatic expressions like "kick the bucket" might need culturally relevant equivalents in other languages to convey the same meaning. Additionally, cultural norms around politeness, humor, and social hierarchy affect how translations should be tailored to resonate appropriately with different audiences.
> > > >
> > > > In summary, the tasks adopted our paper are culture-related, while there are also other broader tasks that are also related to culture. However, it is challenging to perform multi-culture evaluation since most of the tasks do not come in many language/culture, which is why we adopted those detection tasks in the paper. We hope that our work can be one that tries to build such a cultural benchmark and inspire new ones in the future.
> > > >
> > > > - - -
> > > >
> > > > Again, thanks for your professional feedback to our paper to make it even better! If you think our response has addressed your concerns, please increase the rating; otherwise, we are happy to address your further concerns:)

---

> > > > > ### Comment · Reviewer_eAV9 · 2024-08-10
> > > > >
> > > > > Can you also evaluate with a few more tasks? Such as sentiment analysis, irony and sarcasm detection. You don't need to cover all the languages you included in the paper, you can have a few languages to strengthen your results.

---

> > > > > > ### Author Response · Authors · 2024-08-11
> > > > > > **Further Response**
> > > > > >
> > > > > > Per reviewer's request, we conducted more experiments on culture-related tasks, such as sentiment analysis, abuse detection, flame detection and aggressiveness detection. Results are shown in the table below. Our method outperforms GPT-3.5-turbo on a large margin on all tasks. We will append those results into the final version of our paper.
> > > > > >
> > > > > > | Culture | Task            | Dataset          | Metric   | GPT-3.5-turbo | Ours  |
> > > > > > |---------|-----------------|------------------|----------|---------------|-------|
> > > > > > | Arabic  | Sentiment       | ArSAS[1]         | Macro-F1 | 0.55          | 0.632 |
> > > > > > |         | Emotion         | SemEval2018[2]   | JS       | 0.395         | 0.52  |
> > > > > > |         | Harmful content | CT-CWT-22[3]     | F1(POS)  | 0.471         | 0.651 |
> > > > > > |         | Dialect         | QADI[4]          | Macro-F1 | 0.07          | 0.1   |
> > > > > > |  German | Sentiment       | Tweet[5]         | Macro-F1 | 0.562         | 0.63  |
> > > > > > |         | abuse           | GermEval[6]      | Macro-F1 | 0.473         | 0.578 |
> > > > > > |         | flame           | News[7]          | Macro-F1 | 0.512         | 0.56  |
> > > > > > | Spanish | Sentiment       | Tweet[8]         | Macro-F1 | 0.679         | 0.721 |
> > > > > > |         | Misogyny        | Tweet[9]         | Macro-F1 | 0.395         | 0.457 |
> > > > > > |         | Aggressiveness  | mex-a3t[10]      | Macro-F1 | 0.41          | 0.438 |
> > > > > > | Bengali | Sentiment       |        VADER[11] | Macro-F1 | 0.621         | 0.672 |
> > > > > > |         | Aggressiveness  | Youtube[12]      | Macro-F1 | 0.782         | 0.847 |
> > > > > > |         | Misogyny        | Youtube[12]      | Macro-F1 | 0.823         | 0.91  |
> > > > > > | Chinese | Sentiment       | Dictionary[13]   | Macro-F1 | 0.687         | 0.732 |
> > > > > > |         | abusive         | COLD[14]         | Macro-F1 | 0.81          | 0.873 |
> > > > > > |  Korean | Sentiment       | Steam[15]        | Macro-F1 | 0.687         | 0.712 |
> > > > > > |         | Sentiment       | NSMC[16]         | Macro-F1 | 0.83          | 0.851 |
> > > > > >
> > > > > > [1] Elmadany, A. A., H. Mubarak, and W. Magdy. "ArSAS: An Arabic Speech-Act and Sentiment Corpus of Tweets. 2018." Available online: lrec-conf. org/workshops/lrec2018 W. Vol. 30.
> > > > > >
> > > > > > [2] Mohammad, Saif, et al. "Semeval-2018 task 1: Affect in tweets." Proceedings of the 12th international workshop on semantic evaluation. 2018
> > > > > >
> > > > > > [3] Nakov, Preslav, et al. "Overview of the CLEF-2022 CheckThat! lab task 1 on identifying relevant claims in tweets." 2022 Conference and Labs of the Evaluation Forum, CLEF 2022. CEUR Workshop Proceedings (CEUR-WS. org), 2022
> > > > > >
> > > > > > [4] Abdelali, Ahmed, et al. "Arabic dialect identification in the wild." arXiv preprint arXiv:2005.06557 (2020)
> > > > > >
> > > > > > [5] Cieliebak, Mark, et al. "A twitter corpus and benchmark resources for german sentiment analysis." 5th International Workshop on Natural Language Processing for Social Media, Boston MA, USA, 11 December 2017. Association for Computational Linguistics, 2017.
> > > > > >
> > > > > > [6] Wiegand, Michael, Melanie Siegel, and Josef Ruppenhofer. "Overview of the germeval 2018 shared task on the identification of offensive language." (2018): 1-10.
> > > > > >
> > > > > > [7] Steinberger, Josef, et al. "Cross-lingual Flames Detection in News Discussions." RANLP. 2017.
> > > > > >
> > > > > > [8] Paredes-Valverde, Mario Andrés, et al. "Sentiment analysis in Spanish for improvement of products and services: A deep learning approach." Scientific Programming 2017.1 (2017): 1329281.
> > > > > >
> > > > > > [9] Fersini, Elisabetta, Paolo Rosso, and Maria Anzovino. "Overview of the task on automatic misogyny identification at IberEval 2018." Ibereval@ sepln 2150 (2018): 214-228.
> > > > > >
> > > > > > [10] Álvarez-Carmona, Miguel Ángel, et al. "Overview of MEX-A3T at IberEval 2018: Authorship and Aggressiveness Analysis in Mexican Spanish Tweets." IberEval@ SEPLN. 2018.
> > > > > >
> > > > > > [11] Amin, Al, et al. "Bengali vader: A sentiment analysis approach using modified vader." 2019 International Conference on Electrical, Computer and Communication Engineering (ECCE). IEEE, 2019.
> > > > > >
> > > > > > [12] Kumar, Ritesh, et al. "Evaluating aggression identification in social media." Proceedings of the second workshop on trolling, aggression and cyberbullying. 2020.
> > > > > >
> > > > > > [13] Xu, Guixian, et al. "Chinese text sentiment analysis based on extended sentiment dictionary." IEEE access 7 (2019): 43749-43762.
> > > > > >
> > > > > > [14] Deng, Jiawen, et al. "COLD: A benchmark for Chinese offensive language detection." arXiv preprint arXiv:2201.06025 (2022).
> > > > > >
> > > > > > [15] https://github.com/bab2min/corpus/tree/master/sentiment
> > > > > >
> > > > > > [16] https://github.com/e9t/nsmc
> > > > > >
> > > > > > ---
> > > > > >
> > > > > > If those results can address your concerns, please consider raise the rating! We are also happey for further discussion on your concerns. Thanks for your support!

---

> > > > > > > ### Comment · Reviewer_eAV9 · 2024-08-11
> > > > > > >
> > > > > > > Thank you for your new results. Most of my concerns have been addressed. I updated my rating accordingly.
> > > > > > >
> > > > > > > You may find these benchmarks useful for evaluating other more diverse tasks. Please consider using some of these tasks and datasets to evaluate your models in your final version of the paper.
> > > > > > >
> > > > > > >  1. Francesco Barbieri, Luis Espinosa Anke, and Jose Camacho-Collados. 2022. XLM-T: Multilingual Language Models in Twitter for Sentiment Analysis and Beyond. In Proceedings of the Thirteenth Language Resources and Evaluation Conference, pages 258–266, Marseille, France. European Language Resources Association.
> > > > > > > 2. Chiyu Zhang, Khai Doan, Qisheng Liao, and Muhammad Abdul-Mageed. 2023. The Skipped Beat: A Study of Sociopragmatic Understanding in LLMs for 64 Languages. In Proceedings of the 2023 Conference on Empirical Methods in Natural Language Processing, pages 2630–2662, Singapore. Association for Computational Linguistics.

---

> > > > > > > > ### Author Response · Authors · 2024-08-12
> > > > > > > >
> > > > > > > > Thanks for your support! We will conduct more experiments on the benchmarks you mentioned and append into the final version of our paper!

---

### Official Review · Reviewer_XR5a · 2024-07-06

**Soundness:** 2
**Presentation:** 3
**Contribution:** 2
**Rating:** 5
**Confidence:** 4

**Summary:**

This paper introduces CultureLLM, a novel and cost-effective approach to address the cultural biases in Large Language Models (LLMs) that arise from the dominance of English training data. Traditional solutions like prompt engineering and culture-specific pre-training are either expensive or computationally intensive, and often fail to address the paucity of data from low-resource cultures. CultureLLM leverages the World Value Survey (WVS) as seed data and employs a semantic data augmentation method to generate additional training data. The authors utilized only 50 seed samples fromWVS, extending them with augmented data to fine-tune culture-specific LLMs and a unified model, CultureLLM-One, covering 9 diverse cultures, including both rich and low-resource languages.

**Strengths:**

- The paper studies culture understanding which is an important problem.
- The paper proposes an interesting data collection framework through role-playing.

**Weaknesses:**

- __Assumption of Language as Culture__. The paper equates languages with cultures, which is an oversimplification. Cultures are multi-faceted and cannot be fully encapsulated by language alone. There are significant cultural differences within the same language-speaking regions that may not be adequately captured by this approach. Even worse, on line 205, the authors said they picked “representative countries''. But how do you define “representative”? More importantly, this selection method will likely cause the fine-tuned LLMs to be biased towards these “representative” countries for certain languages.

- __Limited Scope of Seed Data__.  The methodology relies heavily on a small seed dataset of only 50 samples from the World Value Survey (WVS). While the approach is cost-effective, it may not capture the full breadth and nuances of cultural diversity.


- __Unfair Baseline Comparison__. The paper claims better performance than other culture-specific LLMs, like TaiwanLLM and SeaLLM. However, the comparison is unfair as these LLMs use different architectures and different amounts of pre-training data. The author could try fine-tuning the gpt-3.5-turbo with these baseline models’ fine-tuning/instruction-tuning data. Until then, “cost-effective” cannot be claimed.

**Questions:**

- I am uncertain about the relevancy between the evaluated tasks and the fine-tuning data. How are values in the World Value Survey/extracted opinions relevant to offensive language/ hate speech/ stance? Can you elaborate?

**Limitations:**

The assumption of language as culture is a significant limitation of this work.

---

> ### Author Rebuttal · Authors · 2024-08-05
>
> **W1: Assumption of Language as Culture. The paper equates languages with cultures, which is an oversimplification. Cultures are multi-faceted and cannot be fully encapsulated by language alone. There are significant cultural differences within the same language-speaking regions that may not be adequately captured by this approach. Even worse, on line 205, the authors said they picked “representative countries''. But how do you define “representative”? More importantly, this selection method will likely cause the fine-tuned LLMs to be biased towards these “representative” countries for certain languages.**
>
> *Why using one language to denote one culture:*
> We strongly agree that language is not equal to, but only a part of culture. But using language to study culture is possible due to the following aspects:
> - Existing literature on culture understanding shows that culture boundaries are fluid, dynamic and uncertain. Delanoy emphasizes that cultures are not homogeneous or static entities but are fluid and dynamic. He critiques essentialist views that rigidly define cultural boundaries and instead promotes a more nuanced understanding that considers the intersections of various cultural factors, such as ethnicity, language, religion, and socio-economic conditions [1]. Appadurai also discusses the fluidity of cultural boundaries and the creation of new cultural forms [2]. Cultural boundaries can be geographical regions, language, religion and so on. Based on above statements, using language as cultural boundaries is reasonable.
> - Existing NLP works on culture also leverage labguage as culture boundaries. [3] focuses on Arabic and English culture. [4] focuses on 8 different cultures: English, Chinese, French, Russian, German, Arabic, Japanese and Korean. [5] also use language to split different cultures. The authors work on English, German, Russian, Bengali, Chinese, and Indonesian culture. [6] is a hand-crafted benchmark for evaluate diverse cultures. They also use languages as culture boundaries.
> - Most downstream benchmarks are classified via language and we cannot get more fine-grained perspectives. For example, if we want to evaluate the performance of Arabic model, we can find benchmarks in Arabic culture. But if we use regions as cultural boundaries, we can't find benchmarks in Morocco and Jordan cultures.
> - It’s interesting to incorporate more fine-grained cultural differences within the same language-speaking regions into LLMs. Our method is an initial attempt, which can generalize to more fine-grained cultures. Just changing the source of seed data can achieve this. However, we can not implement this method for all fine-grained cultures in one paper, because of time and resource limit.
>
> *Explanation on "representative countries":*
>
> In terms of “representative countries”, our selection criterion is that we choose the country which has the *most population*. We agree that this can cause the fine-tuned LLMs to be biased towards the “representative countries”. However, we think the criterion may be the best way to align with majority of people from certain cultures.
>
> Finally, note that the main contribution of the paper is to present a general algorithm that can augment LLM culture data but *not* specific to any cultures or contries. In the future, if more fine-grained culture data are available, our algorithm can also work well.
>
>
> [1] Delanoy, Werner. "What is culture." The Cambridge handbook of intercultural communication (2020): 17-34.
>
> [2] Appadurai, Arjun. Modernity at large: Cultural dimensions of globalization. Vol. 1. U of Minnesota Press, 1996.
>
> [3] Naous, Tarek, et al. "Having beer after prayer? measuring cultural bias in large language models." ACL (2024).
>
> [4] Wang, Wenxuan, et al. "Not all countries celebrate thanksgiving: On the cultural dominance in large language models." arXiv preprint arXiv:2310.12481 (2023).
>
> [5] Liu, Chen Cecilia, et al. "Are multilingual llms culturally-diverse reasoners? an investigation into multicultural proverbs and sayings." arXiv preprint arXiv:2309.08591 (2023).
>
> [6] Myung, Junho, et al. "BLEnD: A Benchmark for LLMs on Everyday Knowledge in Diverse Cultures and Languages." arXiv preprint arXiv:2406.09948 (2024).
>
> **W2: Limited Scope of Seed Data. The methodology relies heavily on a small seed dataset of only 50 samples from the World Value Survey (WVS). While the approach is cost-effective, it may not capture the full breadth and nuances of cultural diversity.**
>
> - First, although the 50 data seem narrow, they however carry resourceful information since they are carefully designed and validaded by human experts. The seed data covers 7 topics: social values, security, science and technology, religious values, ethical values and norms, political interest and political participation, and migration. The data is high quality and have been widely used by a lot of literature. This statement can also be verified by the excellent performance of CultureLLM on downstream tasks.
> - Second, we have conduct pilot experiments on selecting the seed data from WVS. In our study, we explore the effect of sample’s quantity and different composition of samples on the performance of model. The results indicate that the 50 samples perform better than other settings. The reasons may be that fine-tuned LLMs perform better when the distribution of fine-tuned data similar to that of pre-training data. The distribution of those 50 seed data is similar to the pre-training data. So we select those samples.
> - Finally, the *main contribution* of the paper is not the seed data, but our data augmentation algorithm that remains general to any seed data and topics. That being said, in the future, with new topics and seed data available, users can easily develop their own versions of LLMs using our algorithm.
>
> **W3: Unfair Baseline Comparison.[...]Until then, “cost-effective” cannot be claimed.**
>
> We response in general response for this part.

---

> ### Author Response · Authors · 2024-08-05
> **Remaining Rebuttal**
>
> **Q1: I am uncertain about the relevancy between the evaluated tasks and the fine-tuning data. How are values in the World Value Survey/extracted opinions relevant to offensive language/ hate speech/ stance? Can you elaborate?**
>
> We analyze the performance for each task and report the WinRate in the table below.
>
> |            | offensive detect | hate detect | stance detect | toxicity detect | threat detect | bias detect | abusive detect | spam detect |
> | ---------- | ---------------- | ----------- | ------------- | --------------- | ------------- | ----------- | -------------- | ----------- |
> | ChatGPT    | 0.6143           | 0.5433      | 0.6758        | 0.5280          | 0.4270        | 0.4464      | 0.5889         | 0.5846      |
> | CultureLLM | 0.7203           | 0.6197      | 0.7359        | 0.6859          | 0.5172        | 0.5077      | 0.6622         | 0.6451      |
> | WinRate        | 0.1060           | 0.0764      | 0.0600        | 0.1579          | 0.0903        | 0.0612      | 0.0733         | 0.0605      |
>
> The relevance of each task with WVS can be described in the following:
> - offensive language detect:
> 1. Cultural Context and Sensitivity to Offensive Language: The World Values Survey aims to capture cultural values and beliefs across different societies. One aspect of cultural values is the tolerance or acceptance of offensive language. In some cultures, certain words or expressions may be considered highly offensive, while in others they may be more tolerated or even commonly used.
> - hate speech detect:
> 1. Societal Norms and Attitudes: The WVS provides data on societal norms, attitudes towards minorities, and levels of societal trust. This data can help understand the underlying societal conditions that might foster hate speech or, conversely, promote tolerance and inclusivity.
> 2. Cultural Context: Understanding the cultural context is crucial for effectively detecting and interpreting hate speech. The WVS offers a rich dataset for understanding cultural differences in values and norms, which can inform more nuanced hate speech detection algorithms.
> - stance detect:
> 1. Understanding Contextual Influences on Stance: The WVS can provide the cultural and societal background needed to understand why certain stances are more prevalent in specific regions or among certain demographic groups. This context can be invaluable for interpreting the results of stance detection analyses, especially when comparing stances across different cultures and societies.
> - toxicity detect:
> 1. Reflection of Societal Norms in Online Behavior: The WVS provides insights into the prevailing norms and values within societies, which can indirectly inform the context within which toxic behavior manifests online. Understanding societal attitudes towards diversity, authority, individual freedom, and tolerance can help in interpreting the root causes of toxic behavior and devising appropriate responses.
> - threat detect:
> 1. Understanding Motivations and Behaviors: Insights from the WVS can help understand the cultural and societal contexts that may influence the behavior of individuals or groups posing threats. This knowledge can inform more targeted and effective threat detection and mitigation strategies that consider the root causes of conflict or aggression.
> 2. Cultural Sensitivity in Security Measures: Incorporating findings from the WVS can lead to more culturally sensitive security practices that respect local values and norms. This is crucial in global operations where misunderstanding cultural nuances can lead to ineffective or counterproductive security measures.
> - bias detect:
> 1. Understanding Societal Norms and Attitudes: Insights from the WVS can help in understanding the cultural and societal norms that underlie biases. By analyzing patterns in global values and beliefs, we can identify prevalent stereotypes, prejudices, and discriminatory attitudes that may need to be addressed in bias detection efforts
> 2. Injection of More cultural nuances: The WVS data can provide valuable context that are sensitive to cultural differences in values and norms. This is better equipped to detect and mitigate biases in data sets that reflect cultural nuances, ensuring that AI-driven decisions are fair and equitable across different societal contexts.
> - abusive detect:
> 1. Cultural Contexts of Abuse: The WVS can help identify cultural norms that influence perceptions of what constitutes abusive behavior. This is crucial for developing detection systems that are sensitive to cultural differences, ensuring that they can effectively identify abuse without mistakenly flagging culturally specific but non-abusive interactions.
> 2. Injection of More cultural nuances: Insights from the WVS can inform the development of more nuanced algorithms for detecting abusive behavior by providing context on societal values and norms.

---

> ### Comment · Reviewer_XR5a · 2024-08-08
> **Reviewer's response**
>
> While I acknowledge that prior works have used language as a proxy for culture, the validity of this approach remains debatable. Using language as a cultural boundary can simplify the implementation, but it doesn't fully address the complexity and diversity of cultures that share a common language. Regarding the choice of “representative countries,” the authors mentioned, "we think the criterion may be the best way to align with the majority of people from certain cultures." However, isn’t that exactly the flip side of having a biased model?
>
> In terms of the seed data, your explanations have raised more questions. First, in your pilot study, what other settings did you compare these 50 seed data against? Second, it is confusing when you refer to the seed data as being "similar to that of pre-training data." Are you referring to the pre-training data of GPT-3.5-turbo? If so, how do you know the distribution of proprietary data? Your response implies that you already know the pre-training data of GPT-3.5-turbo, which raises concerns about the methodology.
>
> As for your justification regarding fair comparison, there are also some issues. First, referencing other leaderboards does not justify your experimental setup. Second, acknowledging that a fair comparison cannot be achieved due to data unavailability suggests that it is premature to label your approach as "cost-effective." We need to clarify which costs and what effectiveness metrics you are comparing against. Lastly, the mention of “CulturePark” in your response makes it seem like the responses are being reused for multiple submission, which may come across as unprofessional.

---

> > ### Author Response · Authors · 2024-08-09
> > **Further Response**
> >
> > We thank reviewer XR5a for your prompt response to our rebuttal. Now we address your further concerns.
> >
> > > While I acknowledge that prior works have used language as a proxy for culture, the validity of this approach remains debatable.
> >
> > Agreed. We are certainly not the fist work to use language as a proxy for culture and this is not our contribution. We hope that our work is not judged on this point.
> >
> > > Using language as a cultural boundary can simplify the implementation, but it doesn't fully address the complexity and diversity of cultures that share a common language.
> >
> > Agreed. We never claimed such proxy can solve complexity and diversity of cultures. This is beyond the scope of the paper.
> >
> > > Regard "representative countries" [...] isn’t that exactly the flip side of having a biased model?
> >
> > Good point. The ideal state is that we can use the data from both language-rich and language-poor countries. But the bitter reality is that not only us, but also most of the researchers *cannot* make good use of language-poor countries since the *labeled* data remains extremely unavailable. The main point of the paper is not aiming at extremely-low resource culture (but we will be in the future). Furthermore, We would like to point out that in LLM world, *any language other than English* should be treated as "poor" language since the pre-training amount is significantly less than English. In this sense, our contributions can be viewed as extending the cultural understanding ability to non-English, but not-so-poor languages. Extending to the poorer languages could still be a problem.
> >
> > For your comments about biased model: Indeed, bias cannot be overlooked. But our models are less biased than the original ChatGPT on English-dominated models. We will add such discussion in the futher version of the paper.
> >
> > > First, in your pilot study, what other settings did you compare these 50 seed data against?
> >
> > We chose different types of seed data based on different criterion which finally supported us to choose the 50 seed data.
> >
> > In summary, there are 294 questions in World Value Survey. Different seed data can bring benefits at different scales. Our pilot study is to find the best seed data from those 294 questions which can bring more improvement. We randomly select different questions as seed data and evaluate how much they can bring improvement on downsteam tasks. Finally, we select those 50 questions and the corresponding answers as seed data. The table below shows the results of our pilot study in Arabic, Bengali and Chinese cultures. "Avg performance" means the average performance of those three cultures.
> >
> > | Selection criterion          | Avg performance | Min_30.0% Prob  | Min_40.0% Prob  |
> > |------------------------------|-----------------|-----------------|-----------------|
> > | Random selection of 50(1)      |        .4211         | .3846|.4231|
> > | Random selection of 50(2)   |         .4815        | .4322|.4443|
> > | Random selection of 100(1)  |         .4933        |.4622|.4513|
> > | Random selection of 100(2)  |         .4815        | .4312|.4341|
> > | Random selection of 150(1)  |         .5233        |.4722|.4713|
> > | Random selection of 150(2)  |         .5311        |.4722|.4842|
> > | Ours                         |       .5917          |.4832|.4954|
> >
> > > It is confusing when you refer to the seed data as being "similar to that of pre-training data." [...]
> >
> > There is a seminal work focusing on detecting pretraining data of black-box LLMs [1]. Leveraing this work, we tried to explore if the seed data is (probably) trained on GPT-3.5-turbo. We guess that fine-tuned LLMs perform better when the distribution of fine-tuned data similar to that of pre-training data, To verify our hypothesis, we applied this method on those settings and determine if they are in pretraining data. The table above shows the results. "Min_30.0% Prob" and "Min_40.0% Prob" represent the possibility in different settings. The results show that the seed data can bring more improvement when they are more possible to be in the pre-training data. It aligns with our hypothesis. However, we would like to point out that this is just some assumption. We will revise the paper accordingly.
> >
> > [1] Shi, Weijia, et al. "Detecting pretraining data from large language models." ICLR (2024).

---

> > > ### Author Response · Authors · 2024-08-09
> > >
> > > > referencing other leaderboards does not justify your experimental setup
> > >
> > > Agreed. We referenced other leaderboards to show that even the most popular leaderboards in industry and academia cannot guarantee absolute fairness (imagine how competitive they are). It is never easy to do that in LLM era, but all we can do is to try our best to provide relative fairness which we hope that reviewer can understand. If the reviewers thinks more ablations or comparisons are needed to ensure further fairness, we are happy to add them if data and hardware resources are available.
> > >
> > > > Justification of "cost-effective"
> > >
> > > Good question. Now we summarize why our method is "cost-effective":
> > >
> > > - In terms of *money cost*, fine-tuning a language-specific LLM only costs $6, which is extremely cheaper compared to existing models such as SeaLLM and Taiwan LLM.
> > > - In terms of *data cost*, our approach does not need to collect labeled data manually, but only need 50 seed data from WVS (and any future new survey data), which is extremely cheaper compared to those that need heavy data annotation.
> > > - In terms of *time cost*, fine-tuning a language-specific LLM only costs 1-2 hours, which is extremely less than any other cultural specific models which require pre-training and fine-tuning.
> > > - In terms of *effectiveness*, our fine-tuned models can outperform the counterparts with a large margin.
> > > - In terms of *simplicity*, our algorithm is simple, requires only common access to OpenAI API, and provides equitable use to every one.
> > >
> > > In summary, we believe above five perspectives can be used to show that our approach is "cost-effective". We will add such discussion in the future version.
> > >
> > > > Reagarding the term "CulturePark"
> > >
> > > Apoligies for such a mistake.
> > >
> > > - - -
> > >
> > > Again, we thank you for your professional feedback to our paper to make it even better! If you think our response has addressed your concerns, please reconsider the rating; otherwise, we are happy to address your further concerns:)

---

> > > > ### Author Response · Authors · 2024-08-11
> > > >
> > > > Dear reviewer XR5a,
> > > >
> > > > As the discussion phase is about to end and we really tried our best to resolve your concerns, could you please acknowledge if your concerns are addressed? If so, please reconsider the rating; if not, we are very happy to resolve your further concerns. Thank you.
> > > >
> > > > Authors of CultureLLM

---

> ### Comment · Reviewer_XR5a · 2024-08-11
> **Reviewer's response**
>
> I am unsure why the use of language as a proxy to study culture cannot be judged here, as it is a fundamental assumption of your work.
>
> Regarding the selection of the 50 seed data, there seems to be a contradiction. Initially, you mentioned that "the distribution of those 50 seed data is similar to the pre-training data," implying that Shi et al.’s method was used for selection. However, your experiments with settings other than your 50 seed data suggest that these examples were chosen based on average performance. Could you clarify which criteria were ultimately used for the selection?
>
> Moreover, Shi et al.’s method requires computing token probability, which is not available in GPT-3.5-turbo's output. How did you obtain the probability for each token?

---

> > ### Author Response · Authors · 2024-08-12
> > **Further Response**
> >
> > > I am unsure why the use of language as a proxy to study culture cannot be judged here, as it is a fundamental assumption of your work.
> >
> > Indeed, it's an open question without groundtruth answers. On this debatable problem, while we used language as a cultural proxy as suggested by many other works, we also agree and respect that the reviewer may think otherwise. This is not the contribution of our work. We just follow a lot of previous works [1-4].
> >
> >
> > [1] Naous, Tarek, et al. "Having beer after prayer? measuring cultural bias in large language models." ACL (2024).
> >
> > [2] Wang, Wenxuan, et al. "Not all countries celebrate thanksgiving: On the cultural dominance in large language models." arXiv preprint arXiv:2310.12481 (2023).
> >
> > [3] Liu, Chen Cecilia, et al. "Are multilingual llms culturally-diverse reasoners? an investigation into multicultural proverbs and sayings." arXiv preprint arXiv:2309.08591 (2023).
> >
> > [4] Myung, Junho, et al. "BLEnD: A Benchmark for LLMs on Everyday Knowledge in Diverse Cultures and Languages." arXiv preprint arXiv:2406.09948 (2024).
> >
> > > Regarding the selection of the 50 seed data, there seems to be a contradiction. [...] Could you clarify which criteria were ultimately used for the selection?
> >
> >
> > Sorry for the misunderstanding. We chose those 50 seed data based on the performance on downstream tasks. Because of the different performance of seed data, we assumed that "The reasons may be that fine-tuned LLMs perform better when the distribution of fine-tuned data similar to that of pre-training data.". To verify our hypothesis, we applied Shi et al.’s method on our data. The results showed that our hypothesis is reasonable.
> >
> >
> > > Shi et al.’s method requires computing token probability, which is not available in GPT-3.5-turbo's output. How did you obtain the probability for each token?
> >
> > Actually, the token probability of GPT-3.5-turbo is available according to the OpenAI API document [1]:
> >
> >     from openai import OpenAI
> >     client = OpenAI()
> >
> >     completion = client.chat.completions.create(
> >       model="gpt-3.5-turbo-0125",
> >       messages=[
> >         {"role": "user", "content": "Hello!"}
> >       ],
> >       logprobs=True,
> >       top_logprobs=2
> >     )
> >
> >     print(completion.choices[0].message)
> >     print(completion.choices[0].logprobs)
> >
> > Part of the output:
> >
> >     {
> >         ...
> >       "choices": [
> >         {
> >           "index": 0,
> >           "message": {
> >             "role": "assistant",
> >             "content": "Hello! How can I assist you today?"
> >           },
> >           "logprobs": {
> >             "content": [
> >                         {
> >                 "token": "Hello",
> >                 "logprob": -0.31725305,
> >                 "bytes": [72, 101, 108, 108, 111],
> >                 "top_logprobs": [
> >                   {
> >                     "token": "Hello",
> >                     "logprob": -0.31725305,
> >                     "bytes": [72, 101, 108, 108, 111]
> >                   },
> >                   {
> >                     "token": "Hi",
> >                     "logprob": -1.3190403,
> >                     "bytes": [72, 105]
> >                   }
> >                 ]
> >               },
> >               {
> >                 "token": "!",
> >                 "logprob": -0.02380986,
> >                 "bytes": [
> >                   33
> >                 ],
> >                 "top_logprobs": [
> >                   {
> >                     "token": "!",
> >                     "logprob": -0.02380986,
> >                     "bytes": [33]
> >                   },
> >                   {
> >                     "token": " there",
> >                     "logprob": -3.787621,
> >                     "bytes": [32, 116, 104, 101, 114, 101]
> >                   }
> >                 ]
> >               },
> >     }
> >
> >
> > For Shi et al.’s code, it was wrote almost one year ago. So we updated part of the code with new version, such as func `calculatePerplexity_gpt3` in `src/run.py` [2].
> >
> >
> > [1] https://platform.openai.com/docs/api-reference/chat/create
> >
> > [2] https://github.com/swj0419/detect-pretrain-code/blob/main/src/run.py
> >
> >
> > - - -
> >
> > If our response can address your concerns, please consider raise the rating! We are also happey for further discussion on your concerns. Thanks for your support!

---

> ### Comment · Reviewer_XR5a · 2024-08-13
> **Reviewer's response**
>
> Thank you for providing the OpenAI API call.
>
> However, Shi's method requires obtaining the (log) probability of the input prompt (i.e. the input tokens), while the API you shared only provides probabilities for the output tokens.
>
> Additionally, it appears this API call supports only the top 20 highest probability tokens. Could you please clarify how you obtained the minimum 30% and 40% log probabilities in your experiments?

---

> > ### Author Response · Authors · 2024-08-13
> >
> > We thank reviewer XR5a for the detailed comments on Shi's method. Now we answer your further concerns.
> >
> > >However, Shi's method requires obtaining the (log) probability of the input prompt (i.e. the input tokens), while the API you shared only provides probabilities for the output tokens.
> >
> > Because this step is to get the probability of the input token, our strategy is to prompt GPT-3.5-turbo with ```Just Repeat the following instruction: {seed data}```. Then GPT-3.5-turbo can repeat the seed data and output the probability of every input token in seed data.
> >
> > >Additionally, it appears this API call supports only the top 20 highest probability tokens. Could you please clarify how you obtained the minimum 30% and 40% log probabilities in your experiments?
> >
> > There is another parameter *n*. Its can determine: How many chat completion choices to generate for each input message. Note that you will be charged based on the number of generated tokens across all of the choices. Keep n as 1 to minimize costs.
> > We can adjust *n* and *top_logprobs* to get more probability tokens.
> >
> > In fact, Shi's paper has little to do with our work, since it is just used to help us *potentially understand* why the selected 50 seed data are helpful in fine-tuning (such understanding could be wrong, actually). *That paper is not even mentioned in our manuscript.* As we discussed before, we exploited other methods such as different random selection to justify the 50 seed data.
> >
> > We would like to kindly ask the reviewer to evaluate *our* technical contribution instead of the detailed discussion of only a possible explanation using other's work which is not mentioned in our submission. Our key contributions include: CultureLLM, a cost-effective solution to fine-tune culturally-aware LLMs, a data augmentation approach, and strong experimental results on multuple datasets.
> >
> > - - -
> >
> > Since the discussion is about to end in 1 day, we would like to ask if the reviewer is satisfied with our previous responses w.r.t. your other comments and update the rating accordingly. We thank the reviewer for the continuous discussion.

---

> ### Comment · Reviewer_XR5a · 2024-08-13
> **Reviewer's response**
>
> Thank you for your response. The authors have addressed several of my concerns. However, the fundamental assumption of using language to denote culture, along with the issue of unfair comparison, remains challenging to resolve.
>
> Nonetheless, considering the authors’ efforts in their rebuttal, I have decided to increase my rating.

---

> ### Author Response · Authors · 2024-08-13
> **Further Response**
>
> We thank reviewer XR5a for the improved rating. There seems to be two remaining concerns:
>
> > the fundamental assumption of using language to denote culture
>
> As we explained previously, for this open question, we are not the first work to use this assumption. We respect the reviewer's opinion on this point.
>
> > the issue of fair comparison
>
> For this comment, we have answered reviewer XR5a's demand on fine-tuning GPT models on the pre-training data of SeaLLM and Taiwan LLM by stating that their data is not publicly available.
>
> In fact, the major experiments in the paper are conducted under fair comparison: we compare GPT-3.5-turbo with our fine-tuned GPT-3.5-turbo version, ensuring that they are using the same backbone models. If one only cares about absolute performance, we still compare with GPT-4, the most advanced model to date. We hope that the reviewer can acknowledge this.
>
> In the future, with more multilingual data publicly released, we will continue more comparisons using the same backbone models.
>
> - - -
>
> Authors appreciate the multiple rounds of discussion with reviewer XR5a, which makes the paper more sound. We will include all discussion results and analysis into the final version of the paper.

---

### Official Review · Reviewer_tmAM · 2024-07-11

**Soundness:** 3
**Presentation:** 3
**Contribution:** 3
**Rating:** 6
**Confidence:** 4

**Summary:**

The paper presents CultureLLM, a fine-tuned LLM based on GPT 3.5 and fine-tuned on a cultural survey (in English) on 50 survey questions that are increased through semantically aware augmentation.  9 different languages are chosen with geographic choices about which survey to use to represent the languages. The CultureLLM is then tested in different tasks that have different language splits to show that the geographic/language awareness improves the performance than unaware LLMs.

**Strengths:**

This is an interesting problem that faces LLMs and how they are relevant to different locales. The authors work to setup the problem and also highlight their challenges along the way and how they dealt with them. An example is how one deals with limiting the questions and templates for instructions generation and then using augmentation via an LLM but doing a test for relevancy of the generated augmentation. The ablation studies and experiments show improvements of the more geographically language tuned models.

**Weaknesses:**

1. It is important to heavily note that language is not equal to culture and this tends to cause confusion in this paper. Culture is way more complex than language and it might have been easier to call the language splits as culture for writing but this will introduce misunderstandings that will muddy your message.
2. Augmentation with semantic similarity checks is something that has been worked on before for NLP augmentaiton. How do you deal with the challenge that even with high cosine similarity because of BERT embeddings, sentences that have very dissimilar meanings will be counted as similar or pass through your filter.

**Questions:**

1. A questions that comes to mind, that you partly address, is the crosslingual nature of your finetuning, instead of translation. Would not having both even be better? And what would the effect also of using more local language based LLMs be? e.g. LeoLLM for german langauge.
2. Augmentation with semantic similarity checks is something that has been worked on before for NLP augmentaiton. How do you deal with the challenge that even with high cosine similarity because of BERT embeddings, sentences that have very dissimilar meanings will be counted as similar or pass through your filter.

Note: You refer to the crowdsource study as having details including IRB/Ethics information in Appendix E, but this is not so. You just describe the task questions. Was IRB/Ethics approval sought given that the nature of the questions may for some people find offensive?

**Limitations:**

The authors have worked to highlight their limitations and also challenges with the approach.

---

> ### Author Rebuttal · Authors · 2024-08-05
>
> **W1: It is important to heavily note that language is not equal to culture and this tends to cause confusion in this paper. Culture is way more complex than language and it might have been easier to call the language splits as culture for writing but this will introduce misunderstandings that will muddy your message.**
>
> We strongly agree that language is not equal to, but only a part of culture. But using language to study culture is possible due to the following aspects:
> - Existing literature on culture understanding shows that culture boundaries are fluid, dynamic and uncertain. Delanoy emphasizes that cultures are not homogeneous or static entities but are fluid and dynamic. He critiques essentialist views that rigidly define cultural boundaries and instead promotes a more nuanced understanding that considers the intersections of various cultural factors, such as ethnicity, language, religion, and socio-economic conditions [1]. Appadurai also discusses the fluidity of cultural boundaries and the creation of new cultural forms [2]. Cultural boundaries can be geographical regions, language, religion and so on. Based on above statements, using language as cultural boundaries is reasonable.
> - Existing NLP works on culture also leverage labguage as culture boundaries. [3] focuses on Arabic and English culture. [4] focuses on 8 different cultures: English, Chinese, French, Russian, German, Arabic, Japanese and Korean. [5] also use language to split different cultures. The authors work on English, German, Russian, Bengali, Chinese, and Indonesian culture. [6] is a hand-crafted benchmark for evaluate diverse cultures. They also use languages as culture boundaries.
> - Most downstream benchmarks are classified via language and we cannot get more fine-grained perspectives. For example, if we want to evaluate the performance of Arabic model, we can find benchmarks in Arabic culture. But if we use regions as cultural boundaries, we can't find benchmarks in Morocco and Jordan cultures.
> - Note that the main contribution of the paper is to present a general algorithm that can augment LLM culture data but not specific to any cultures. In the future, if more fine-grained culture data are available, our algorithm can also work well.
>
> [1] Delanoy, Werner. "What is culture." The Cambridge handbook of intercultural communication (2020): 17-34.
>
> [2] Appadurai, Arjun. Modernity at large: Cultural dimensions of globalization. Vol. 1. U of Minnesota Press, 1996.
>
> [3] Naous, Tarek, et al. "Having beer after prayer? measuring cultural bias in large language models." ACL (2024).
>
> [4] Wang, Wenxuan, et al. "Not all countries celebrate thanksgiving: On the cultural dominance in large language models." arXiv preprint arXiv:2310.12481 (2023).
>
> [5] Liu, Chen Cecilia, et al. "Are multilingual llms culturally-diverse reasoners? an investigation into multicultural proverbs and sayings." arXiv preprint arXiv:2309.08591 (2023).
>
> [6] Myung, Junho, et al. "BLEnD: A Benchmark for LLMs on Everyday Knowledge in Diverse Cultures and Languages." arXiv preprint arXiv:2406.09948 (2024).
>
> **W2: Augmentation with semantic similarity checks is something that has been worked on before for NLP augmentaiton. How do you deal with the challenge that even with high cosine similarity because of BERT embeddings, sentences that have very dissimilar meanings will be counted as similar or pass through your filter.**
>
> There are two tricks that work in our study
> - Threshold Tuning and Filtering: Adjust the similarity threshold to balance between false positives and false negatives. We choose a more conservative threshold, which can reduce the likelihood of accepting semantically dissimilar sentences.
> - Use of Cultural Contextual Information: BERT embeddings are context-sensitive, but sometimes context nuances are not fully captured. To mitigate this, we consider incorporating more cultural context.
>
> **Q1: A questions that comes to mind, that you partly address, is the crosslingual nature of your finetuning, instead of translation. Would not having both even be better? And what would the effect also of using more local language based LLMs be? e.g. LeoLLM for german langauge.**
>
> Nice question! We fine-tune LeoLLM with our generated data in both English and German version. Results are shown below:
> | Task                | hate_check | hate_iwg_1 | hate | hate_off | offensive_eval |
> |---------------------|------------|------------|------|----------|----------------|
> | LeoLLM              | 0.46       | 0.21       | 0.18 | 0.33     | 0.42           |
> | LeoLLM+English data | 0.50       | 0.26       | 0.25 | 0.40     | 0.46           |
> | LeoLLM+German data  | 0.51       | 0.32       | 0.27 | 0.42     | 0.50           |
>
> The results indicate that the fine-tuned model perform better when the language of fine-tuned data is same with that of pre-training data. This could be due to the fact that English is much more than the local languages, which can naturally assist fine-tuning.
>
> **Q2: Augmentation with semantic similarity checks is something that has been worked on before for NLP augmentaiton. How do you deal with the challenge that even with high cosine similarity because of BERT embeddings, sentences that have very dissimilar meanings will be counted as similar or pass through your filter.**
>
> The response is shown in the response of W2.
>
> **Note: You refer to the crowdsource study as having details including IRB/Ethics information in Appendix E, but this is not so. You just describe the task questions. Was IRB/Ethics approval sought given that the nature of the questions may for some people find offensive?**
>
> Thanks for the reminder! We have gotten IRB/Ethics approval for the human study.

---

### Official Review · Reviewer_ASzm · 2024-07-11

**Soundness:** 3
**Presentation:** 3
**Contribution:** 3
**Rating:** 7
**Confidence:** 4

**Summary:**

This research addresses cultural bias in large language models (LLMs) caused by training on mostly English data. Existing solutions can be expensive or require a lot of computing power. Here, they propose CultureLLM, a method that uses existing cultural surveys to create more training data and fine-tune LLMs. This method is shown to be effective and efficient for improving the cultural awareness of LLMs for 9 cultures.

**Strengths:**

- The paper is clearly written and easy to follow
- The topic of the paper (culture LLMs) addresses a timely and important issue with the current LLM situation.
- The approach (especially using existing survey data) is simple and reasonable.
- The experiments show the efficacy of the cultureLLM suggested.

**Weaknesses:**

- I don't see major weaknesses

**Questions:**

- I'm wondering how the performance (or errors) are associated with the coverage of the 50 questions used for fine-tuning.

**Limitations:**

I think the limitations brought by the authors are adequate.

---

> ### Author Rebuttal · Authors · 2024-08-05
>
> **Q1: I'm wondering how the performance (or errors) are associated with the coverage of the 50 questions used for fine-tuning.**
>
> We analyze the performance for each task and report the WinRate in the table below.
>
> |            | offensive detect | hate detect | stance detect | toxicity detect | threat detect | bias detect | abusive detect | spam detect |
> | ---------- | ---------------- | ----------- | ------------- | --------------- | ------------- | ----------- | -------------- | ----------- |
> | ChatGPT    | 0.6143           | 0.5433      | 0.6758        | 0.5280          | 0.4270        | 0.4464      | 0.5889         | 0.5846      |
> | CultureLLM | 0.7203           | 0.6197      | 0.7359        | 0.6859          | 0.5172        | 0.5077      | 0.6622         | 0.6451      |
> | WinRate        | 0.1060           | 0.0764      | 0.0600        | 0.1579          | 0.0903        | 0.0612      | 0.0733         | 0.0605      |
>
> The relevance of each task with WVS can be described in the following:
> - offensive language detect:
> 1. Cultural Context and Sensitivity to Offensive Language: The World Values Survey aims to capture cultural values and beliefs across different societies. One aspect of cultural values is the tolerance or acceptance of offensive language. In some cultures, certain words or expressions may be considered highly offensive, while in others they may be more tolerated or even commonly used.
> - hate speech detect:
> 1. Societal Norms and Attitudes: The WVS provides data on societal norms, attitudes towards minorities, and levels of societal trust. This data can help understand the underlying societal conditions that might foster hate speech or, conversely, promote tolerance and inclusivity.
> 2. Cultural Context: Understanding the cultural context is crucial for effectively detecting and interpreting hate speech. The WVS offers a rich dataset for understanding cultural differences in values and norms, which can inform more nuanced hate speech detection algorithms that are sensitive to context and do not inadvertently suppress legitimate expressions of cultural or political dissent.
> - stance detect:
> 1. Understanding Contextual Influences on Stance: The WVS can provide the cultural and societal background needed to understand why certain stances are more prevalent in specific regions or among certain demographic groups. This context can be invaluable for interpreting the results of stance detection analyses, especially when comparing stances across different cultures and societies.
> - toxicity detect:
> 1. Reflection of Societal Norms in Online Behavior: The WVS provides insights into the prevailing norms and values within societies, which can indirectly inform the context within which toxic behavior manifests online. Understanding societal attitudes towards diversity, authority, individual freedom, and tolerance can help in interpreting the root causes of toxic behavior and devising appropriate responses.
> - threat detect:
> 1. Understanding Motivations and Behaviors: Insights from the WVS can help understand the cultural and societal contexts that may influence the behavior of individuals or groups posing threats. This knowledge can inform more targeted and effective threat detection and mitigation strategies that consider the root causes of conflict or aggression.
> 2. Cultural Sensitivity in Security Measures: Incorporating findings from the WVS can lead to more culturally sensitive security practices that respect local values and norms. This is crucial in global operations where misunderstanding cultural nuances can lead to ineffective or counterproductive security measures.
> - bias detect:
> 1. Understanding Societal Norms and Attitudes: Insights from the WVS can help in understanding the cultural and societal norms that underlie biases. By analyzing patterns in global values and beliefs, we can identify prevalent stereotypes, prejudices, and discriminatory attitudes that may need to be addressed in bias detection efforts
> - abusive detect:
> 1. Cultural Contexts of Abuse: The WVS can help identify cultural norms that influence perceptions of what constitutes abusive behavior. This is crucial for developing detection systems that are sensitive to cultural differences, ensuring that they can effectively identify abuse without mistakenly flagging culturally specific but non-abusive interactions.
> 2. Injection of More cultural nuances: Insights from the WVS can inform the development of more nuanced algorithms for detecting abusive behavior by providing context on societal values and norms.
> 4. Evaluating Tolerance Levels: The WVS data can provide insights into societal tolerance levels towards different forms of behavior, including what might be considered abusive. This can help in assessing the urgency and type of interventions needed to address abusive behaviors in various cultural contexts.
> - spam detect
> 1. Cultural Variations in Communication: The WVS can shed light on cultural differences in communication styles and preferences, which can inform more nuanced spam detection algorithms that are better able to distinguish between legitimate mass communications and spam in different cultural contexts.
> 2. Attitudes Towards Technology and Privacy: Insights from the WVS regarding societal attitudes towards technology use, privacy, and data protection can help in tailoring spam detection efforts to respect cultural norms and expectations. For instance, societies with a high value on privacy might be more receptive to stringent spam filters.
> 3. Attitudes Towards Technology and Privacy: Insights from the WVS regarding societal attitudes towards technology use, privacy, and data protection can help in tailoring spam detection efforts to respect cultural norms and expectations. For instance, societies with a high value on privacy might be more receptive to stringent spam filters.

---

> > ### Comment · Reviewer_ASzm · 2024-08-09
> >
> > Thank you for the author response.

---

### Author Rebuttal · Authors · 2024-08-05

Dear Reviewers and AC,

We want to thank all reviewers for pointing out our strengths, including:
- problem significance: "addresses a timely and important issue with the current LLM situation", "an interesting problem that faces LLMs and how they are relevant to different locales"
- novel method: "is simple and reasonable"
- solid experiment: the experiments show the efficacy of the cultureLLM suggested, with solid experiments to investigate the effectiveness of the proposed method
- writing: "well-organized and easy to read"

Rebuttal for each reviewer has been submitted in respective sections. Here, we would like to clarify two common weaknesses.

Specifically, as raised by reviewer *XR5a* and *tmAM*, there remains one common weakness about the relationship between culture and language, which we aim to address here:

We strongly agree that language is *not* equal to, but only *a part of* culture. But using language to study culture is possible due to the following aspects:
- Existing literature on culture understanding shows that culture boundaries are fluid, dynamic and uncertain. Delanoy emphasizes that cultures are not homogeneous or static entities but are fluid and dynamic. He critiques essentialist views that rigidly define cultural boundaries and instead promotes a more nuanced understanding that considers the intersections of various cultural factors, such as ethnicity, language, religion, and socio-economic conditions [1]. Appadurai also discusses the fluidity of cultural boundaries and the creation of new cultural forms [2]. Cultural boundaries can be geographical regions, language, religion and so on. Based on above statements, using language as cultural boundaries is reasonable.
- Existing NLP works on culture also leverage language as culture boundaries. [3] focuses on Arabic and English culture. [4] focuses on 8 different cultures: English, Chinese, French, Russian, German, Arabic, Japanese and Korean. [5] also use language to split different cultures. The authors work on English, German, Russian, Bengali, Chinese, and Indonesian culture. [6] is a hand-crafted benchmark for evaluate diverse cultures. They also use languages as culture boundaries.
- Most downstream benchmarks are classified via language and we cannot get more fine-grained perspectives. For example, if we want to evaluate the performance of Arabic model, we can find benchmarks in Arabic culture. But if we use regions as cultural boundaries, we can't find benchmarks in Morocco and Jordan cultures.
- Note that the main contribution of the paper is to present a general algorithm that can augment LLM culture data but not specific to any cultures. In the future, if more fine-grained culture data are available, our algorithm can also work well.

Another thing we want to discuss is about *"fair comparation"*. We agree that it is necessary to fine-tune gpt models on the training data of TaiwanLLM and SeaLLM. However, a sad story is that their training data is not publicly accessible. In fact, the popular LLM leaderboards such as Chatbot Arena and AlpacaEval ranks models regardless of their sizes, training data, and post-training, but only the final performance on the same benchmarks. Moreover, we realize that it is never easy to reach a "fair" comparison since if we fine-tune the same models on their data, it is unfair for our approach since their pre-training data is significantly larger than ours. We would like to claim that given limited budget and GPU hardware, CulturePark remains a cost-effective solution to fastly build a cultural-specific LLM for low-resource culture. This is the main contribution of the paper.


[1] Delanoy, Werner. "What is culture." The Cambridge handbook of intercultural communication (2020): 17-34.

[2] Appadurai, Arjun. Modernity at large: Cultural dimensions of globalization. Vol. 1. U of Minnesota Press, 1996.

[3] Naous, Tarek, et al. "Having beer after prayer? measuring cultural bias in large language models." ACL (2024).

[4] Wang, Wenxuan, et al. "Not all countries celebrate thanksgiving: On the cultural dominance in large language models." arXiv preprint arXiv:2310.12481 (2023).

[5] Liu, Chen Cecilia, et al. "Are multilingual llms culturally-diverse reasoners? an investigation into multicultural proverbs and sayings." arXiv preprint arXiv:2309.08591 (2023).

[6] Myung, Junho, et al. "BLEnD: A Benchmark for LLMs on Everyday Knowledge in Diverse Cultures and Languages." arXiv preprint arXiv:2406.09948 (2024).

- - -

We hope that your concerns can be addressed. Thank you for your hard work.

Authors of CultureLLM

---

### Decision · Program_Chairs · 2024-09-25

**Decision:**

Accept (poster)

**Comment:**

This paper proposed CultureLLM, a novel cost-effective approach to addressing culture bias in LLMs. Since prompt engineering and culture-specific pre-training are expensive, CultureLLMs leverages surveys as seed data to further augment the training data and fine-tune the LLMs. The model shows improvements on LLM cultural awareness for 9 cultures.

Strengths:
Overall, the paper is clearly written, easy to follow and presents a novel cost-effective approach to address cultural biases in LLMs called CultureLLM. Authors have interesting way to setup the problem, they leveraged surveys to generate additional training data and fine-tune culture specific LLM and a unified model.

Weaknesses:
There are no major weaknesses. Just prior to the final submission, authors should address the questions of the reviewers, specifically
(1) motivate the assumption of language as a culture
(2) explain why evaluation was done on downstream tasks such as offensive language, toxicity’s etc and results may be biased towards such task; address the question “unfair baseline comparisons” and also are there tasks which could better capture culture-related task
(3) address the feedback on “unfair baseline comparison” with other LLMs given the